# Support-tuned iridium reconstruction with crystalline phase dominating acidic oxygen evolution

Kexin Zhang[1,4], Xiao Liang [1,4], Yucheng Wang [2], Yongcun Zou[1], Xiao Zhao [3], Hui Chen [1] ✉ & Xiaoxin Zou [1] ✉

The dynamic reconstruction of oxygen evolution electrocatalysts dictates their performance, yet conventional Ir-based materials face an inherent activity-stability trade-off due to surface amorphization into hydrous $IrO_x$ phases accompanied by lattice oxygen mechanisms. Here, we uncover a distinct reconstruction pathway for supported Ir nanoparticles, where a $TiO_x$@Ti substrate drives a bulk phase transition from metallic Ir to crystalline rutile $IrO_2$ during electrocatalysis. Unlike surface-limited amorphization, this support-guided crystallization shifts the reaction mechanism from involving lattice oxygen mechanism to the complete adsorbate evolution mechanism, as confirmed by mechanistic and structural analyses. Consequently, the Ir/$TiO_x$@Ti catalyst achieves both high activity and durability in acidic media, demonstrated in three-electrode systems and proton exchange membrane water electrolyzers. This work redefines support roles in electrocatalyst reconstruction, demonstrating that bulk phase engineering—rather than surface modification—resolves the long-standing efficiency-durability conflict in acidic oxygen evolution.

The development of sustainable electrical-to-chemical energy conversion technologies, such as water splitting and $CO_2$ electroreduction, relies fundamentally on efficient anodic oxygen evolution reaction (OER) electrocatalysts[1–3]. Over the past two decades, extensive research has advanced OER catalyst design through mechanistic insights (e.g., scaling relations, reaction descriptors)[4–6], nanostructural engineering (e.g., defect tailoring, heterojunctions)[7–9], and innovative synthesis methods[10]. However, these meticulously designed pre-electrocatalysts—whether oxide or non-oxide—rarely retain their pristine structures under harsh anodic conditions. Instead, they undergo dynamic reconstruction processes driven by electrochemical potentials and electrolyte interactions, leading to irreversible morphological, compositional, or phase transformations[11–13]. The resulting reconstructed phases, rather than the as-synthesized materials,

ultimately govern the catalytic performance. Consequently, deciphering reconstruction pathways and steering the final active phases have emerged as central challenges in OER electrocatalysis, particularly given the unpredictable nature of electrochemical restructuring.

Iridium-based materials stand as the state-of-the-art catalysts for acidic OER, striking a delicate balance between activity and stability[14–16]. To maximize the utilization of this scarce noble metal, diverse strategies—including the synthesis of ultrathin Ir nanocrystals[17–19], Ir nanoalloys[20–22] and supported Ir nanoparticles[23–25] have been explored. The reconstruction chemistry of Ir catalysts, however, has long been a defining limitation. As early as the 1970s, pioneering studies observed that metallic Ir nanoparticles spontaneously form a thin hydrous iridium oxide $IrO_x$ layer on their surfaces during OER[26,27]. Over subsequent decades, this oxidation-driven

[1]State Key Laboratory of Inorganic Synthesis and Preparative Chemistry, College of Chemistry, Jilin University, Changchun, China. [2]State Key Laboratory of Physical Chemistry of Solids, College of Chemistry and Chemical Engineering, Xiamen University, Xiamen, China. [3]Key Laboratory of Automobile Materials of MOE, School of Materials Science and Engineering, Jilin University, Changchun, China. [4]These authors contributed equally: Kexin Zhang, Xiao Liang. ✉e-mail: chenhui@jlu.edu.cn; xxzou@jlu.edu.cn

surface reconstruction was recognized as a ubiquitous phenomenon in Ir-catalyzed OER[28–30]. Although the atomic structure of electrochemically formed IrO$_x$ varies slightly depending on the initial catalyst's crystallinity, defect density and support types[31,32], it is universally described as a disordered, hydrous phase with short-range rutile-like order (Fig. 1a)[33]. While this amorphous IrO$_x$ shell enhances activity via lattice oxygen-mediated pathways (LOM), its metastable nature, characterized by defect-rich and loosely coordinated structure, drives accelerated degradation and Ir dissolution through oxygen vacancy accumulation and structural collapse[34–38]. This intrinsic instability has perpetuated a vexing trade-off: higher activity comes at the cost of reduced durability, hindering the development of practically viable Ir-based OER catalysts.

In this work, we redefine the reconstruction paradigm for Ir electrocatalysts by exploiting support-induced bulk phase engineering. We demonstrate that dispersing Ir nanoparticles on a shell@core TiO$_x$@Ti substrate fundamentally redirects their reconstruction pathway through oxygen dissolution and accumulation in iridium lattices. Instead of the conventional surface-limited amorphization into hydrous IrO$_x$, the catalyst undergoes a complete bulk phase transition from metallic Ir to thermodynamically stable crystalline rutile IrO$_2$ under OER conditions (Fig. 1b). Through a combination of operando X-ray absorption spectroscopy, cyclic voltammetry, isotopic labeling experiments and theoretical calculations, we reveal that this support-guided crystallization shifts the OER mechanism from LOM-participated to the complete AEM. The transition suppresses lattice oxygen participation—a key degradation driver in amorphous IrO$_x$—while maintaining high catalytic activity. Our work positions catalyst supports not merely as static carriers but as catalytic phase regulators, offering a universal strategy to decouple activity and stability in electrochemical systems.

## Results

### Synthesis and structural elucidation of multilayered Ir/TiO$_x$@Ti nanoparticles

The Ir/TiO$_x$@Ti samples, comprising iridium nanoparticles (Ir NPs) supported on shell@core TiO$_x$@Ti particles, were synthesized via a one-pot reaction in ethylene glycol (EG) using titanium nanospheres and H$_2$IrCl$_6$ as precursors (Fig. 2a). Ethylene glycol serves dual roles: (1) reducing [IrCl$_6$]$^{2-}$ to metallic Ir NPs, and (2) mediating the oxidative formation of the shell@core TiO$_x$@Ti support through reaction with dissolved oxygen (Supplementary Figs. 1–6). Raman spectroscopy and X-ray atomic pair distribution function (PDF) analyses reveal that the amorphous TiO$_x$ shell consists of Ti$_3$O$_5$/Ti$_4$O$_7$-like clusters, distinct from thermodynamically stable TiO$_2$ polymorphs (Supplementary Fig. 7).

By varying the H$_2$IrCl$_6$ precursor ratio, we synthesized Ir/TiO$_x$@Ti samples with Ir loadings ranging from 9 to 31 wt%. Powder X-ray diffraction (XRD, Supplementary Fig. 8) and transmission electron microscopy (TEM, Supplementary Fig. 9) results confirm that Ir NPs remain uniformly dispersed below 24 wt% Ir, beyond which Ir shell thickening and nanoparticle agglomeration occur. The 24 wt% Ir/TiO$_x$@Ti sample exhibits optimal mass activity and is selected for subsequent studies. Unsupported Ir NPs, synthesized without titanium nanospheres, serve as a control with a particle size of ~2.1 nm (Supplementary Figs. 10 and 11).

TEM image (Fig. 2b) and scanning electron microscopy image (SEM, Supplementary Fig. 12) reveal that Ir/TiO$_x$@Ti contains nanospheres of ~60 nm with uniformly dispersed Ir NPs on their surfaces. The corresponding elemental mapping (Fig. 2c) shows that the Ti element is more concentrated at the core region of Ir/TiO$_x$@Ti, while the Ir and O elements are enriched at its shell region. The distribution area of each element from the outside gradually decreases in the order of Ir, O and Ti, demonstrating the formation of Ir/TiO$_x$/Ti three-layer shell-core structure. The microstructure of Ir/TiO$_x$@Ti is further characterized using aberration-corrected high-angle annular dark-field scanning TEM (HAADF-STEM). As shown in Fig. 2d, Ir/TiO$_x$@Ti comprises of an Ir nanoparticle layer with a thickness of ~2 nm on the outermost shell, an amorphous TiO$_x$ layer of about 5 nm in the middle, and a single-crystal titanium core. The Ir NPs with a particle size of 2.0 nm are continuous and uniformly distributed, and form a single nanoparticle layer on the amorphous TiO$_x$ overlayer (Fig. 2e). The corresponding line scan results (Supplementary Fig. 13) further show that the iridium signal is concentrated in the outermost 2 nm range,

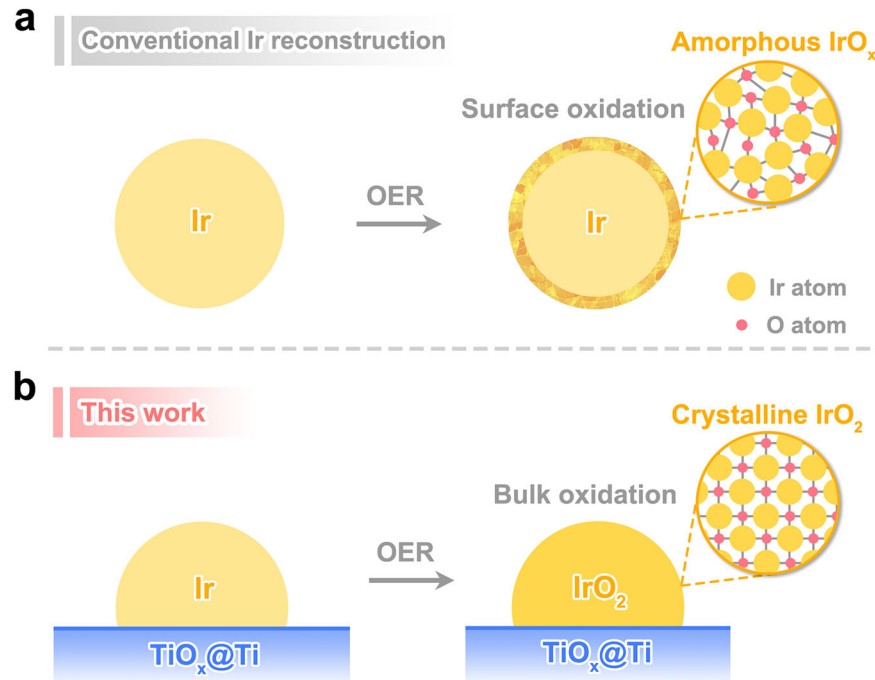

**Fig. 1 | Schematic diagram of electrochemically induced reconstruction of Ir nanoparticles. a** Conventional understanding on the electrochemical oxidation of Ir nanoparticles during OER, which does not depend on the support materials. **b** Extraordinary reconstruction of supported Ir nanoparticles in this work.

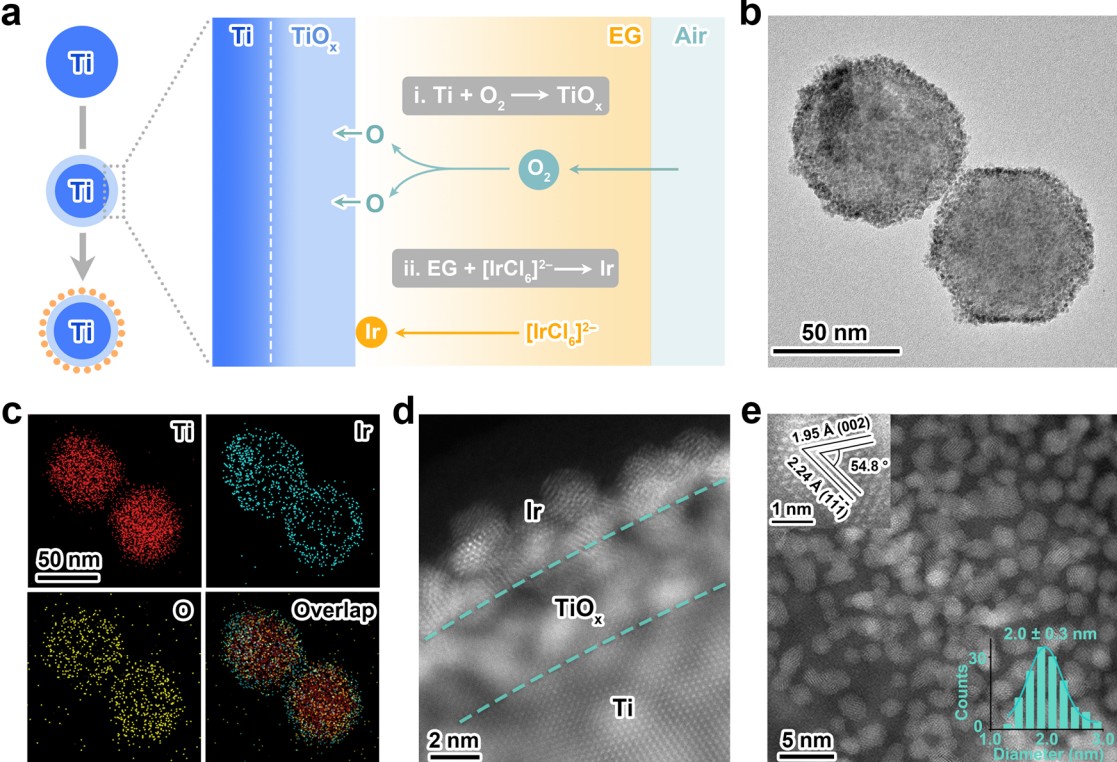

**Fig. 2 | Synthesis and structural characterization of Ir/TiO$_x$@Ti. a** Schematic illustration of the synthesis process for Ir/TiO$_x$@Ti. **b** The TEM image of Ir/TiO$_x$@Ti. **c** The elemental mapping images of Ir/TiO$_x$@Ti. **d**, **e** The aberration-corrected HAADF-STEM images of Ir/TiO$_x$@Ti. **e** The upper left inset shows the lattice fringes of a single iridium nanoparticle, while the lower right inset displays the size distribution histogram obtained from 140 particles.

consistent with the HAADF-STEM observation. Lattice fringes with d-spacings of 1.95 Å and 2.24 Å are observed in the high-resolution HAADF-STEM image (Fig. 2e, inset), which could be ascribed to the (002) and the (1$\bar{1}\bar{1}$) planes of Ir. The dihedral angle between the two is determined as 54.8°, consistent with the theoretical value.

## Catalytic activity and stability of Ir/TiO$_x$@Ti for acidic OER

The OER performance of Ir/TiO$_x$@Ti catalysts with varying Ir loadings (wt%) was systematically evaluated in 0.1 M HClO$_4$ using a three-electrode system (Fig. 3a, inset). As shown in Fig. 3a and Supplementary Fig. 14, geometric activity (current normalized by electrode area) exhibits a volcano-shaped dependence on Ir content, peaking at 24 wt% Ir/TiO$_x$@Ti. This optimal sample demonstrates a mass activity of 192 A/g$_{Ir}$ at 1.53 V versus reversible hydrogen electrode (RHE, Fig. 3b), 2.4-fold higher than unsupported Ir nanoparticles (81 A/g$_{Ir}$). The activity trend correlates directly with Ir utilization efficiency (Fig. 3b, inset): At <24 wt%, incomplete TiO$_x$@Ti surface coverage limits active site exposure; at >24 wt%, Ir nanoparticle agglomeration shields internal particles, reducing accessible active sites.

To probe the structural stability under harsh acidic OER conditions, inductively coupled plasma optical emission spectroscopy (ICP-OES) was employed to quantify Ir dissolution during electrolysis. As shown in Fig. 3c, Ir/TiO$_x$@Ti exhibits a rapid initial dissolution phase (0–2 h) followed by stabilization (2–10 h, total dissolution: 0.16 mg/L), whereas unsupported Ir NPs show continuous dissolution over 8 h (total dissolution: 0.61 mg/L). This 3.8-fold reduction in Ir leaching highlights the TiO$_x$@Ti support's critical role in anchoring Ir NPs and suppressing corrosion. Long-term chronopotentiometric testing at 10 mA/cm$^2$ further reveals durability. Ir/TiO$_x$@Ti undergoes an initial 10-hour activation phase, likely due to surface reconstruction into another active phase, and subsequently maintains stable operation for over 1700 h without degradation (Fig. 3d and Supplementary Fig. 15).

In contrast, unsupported Ir NPs gradually loses their initial activity within 40 h. Notably, Ir/TiO$_x$@Ti's stability outperforms the most recent reports of supported Ir catalysts (typically <200 h, Supplementary Table 1)[23,25,39–42], including TiO$_2$-supported Ir (100 h), Ir/Nb$_2$O$_{5-x}$ (100 h), and Ir/Sb-SnO$_2$ (15 h). These results collectively establish the TiO$_x$@Ti support's dual function: (1) maximizing Ir utilization via optimized nanoparticle dispersion and (2) mitigating Ir dissolution.

## Structural evolution of supported Ir nanoparticles during acidic OER

The notable stability of Ir/TiO$_x$@Ti during prolonged oxygen evolution prompted a systematic investigation of its structural evolution under operational conditions. To elucidate potential-dependent surface transformations, we initially constructed a density functional theory (DFT)-derived surface phase diagram for metallic iridium in acidic media across anodic potentials (Fig. 4a and Supplementary Data 1). At potentials below 0.08 V, the iridium surface forms Ir–H* adsorbates due to strong hydrogen underpotential deposition (H$_{UPD}$), analogous to Pt-group metal behavior in acidic environments[43]. As the potential exceeds 0.08 V, a mixed adsorption regime emerges where HO* and H$_2$O* coexist thermodynamically. Beyond 0.53 V, the surface transitions to complete HO* coverage, ultimately evolving into an O*-terminated structure at potentials above 0.66 V. These computational insights suggest a sequential surface transformation pathway on metallic Ir: Ir–H* → HO*/H$_2$O* → HO* → O*, driven by progressively oxidizing potentials.

We systematically investigated the electrochemical surface oxidation processes of Ir/TiO$_x$@Ti and unsupported Ir nanoparticles using cyclic voltammetry (CV) in 0.1 M HClO$_4$ under anodic conditions. The CV protocol initiated at 0 V vs. RHE, with the upper potential limit (UPL) incrementally increased from 0.6 V to 1.3 V to track

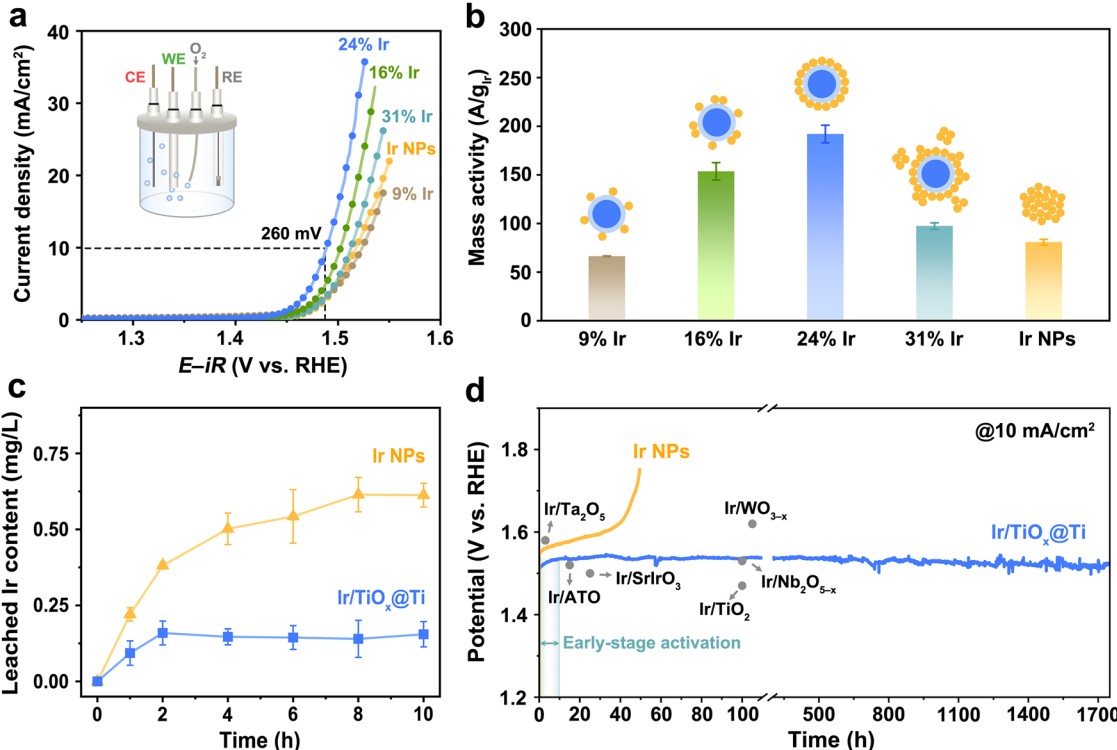

**Fig. 3 | OER performance in three-electrode cell. a** The polarization curves for OER of Ir/TiO$_x$@Ti with varying Ir contents and unsupported Ir NPs with 85% *iR*-correction (0.071 cm$^2$ glassy carbon electrode, 0.18 mg/cm$^2$ Ir loading, 1 mV/s scan rate). The compensation resistances are 39.7 ± 1.2 Ω (9 wt% Ir), 32.9 ± 0.2 Ω (16 wt% Ir), 33.1 ± 0.6 Ω (24 wt% Ir), 33.1 ± 0.1 Ω (31 wt% Ir) and 35.1 ± 0.5 Ω (Ir NPs). The inset shows a schematic diagram of a three-electrode cell. **b** Ir mass activity of Ir/TiO$_x$@Ti and Ir NPs at 1.53 V vs. RHE. The inset shows schematic illustrations depicting the morphological characteristics of Ir/TiO$_x$@Ti and Ir NPs. **c** The content of Ir leached into the electrolyte in the presence of Ir/TiO$_x$@Ti and Ir NPs as the electrocatalysts during the 10-h OER catalysis at 1.55 V vs. RHE without *iR*-compensation. **b**, **c** Error bars are drawn based on the standard deviations of three measurements. **d** The chronopotentiometry curves of Ir/TiO$_x$@Ti and Ir NPs at a current density of 10 mA/cm$^2$ without *iR*-compensation.

oxidation-induced changes (Fig. 4b, c). A distinct peak at -0.15 V, attributed to H$_{UPD}$ on metallic Ir surfaces[44,45], serves as a fingerprint for surface metallic character. For Ir/TiO$_x$@Ti, the H$_{UPD}$ peak intensity decreases above 0.8 V and vanishes completely at 1.1 V (Fig. 4b), indicating irreversible full surface oxidation (Ir–H* → Ir–O* transition). In contrast, unsupported Ir NPs show obvious decrease in H$_{UPD}$ peak intensity above 1.0 V and retain partial H$_{UPD}$ signatures until 1.3 V (Fig. 4c), demonstrating delayed surface oxidation. This 200 mV lower oxidation threshold for Ir/TiO$_x$@Ti highlights the support's role in destabilizing metallic Ir surfaces.

To quantify oxidation kinetics, we performed additional CV cycling (0–1.45 V, 50 mV/s). Ir/TiO$_x$@Ti loses its metallic surface characteristics after just two cycles (H$_{UPD}$ disappearance in Fig. 4d), whereas unsupported Ir NPs require ten cycles to achieve full oxidation (Fig. 4e). This accelerated oxidation kinetics—5-fold faster for Ir/TiO$_x$@Ti—confirms the TiO$_x$@Ti support's ability to lower the activation barrier for Ir surface oxidation, likely through electronic metal-support interactions that facilitate charge transfer during OER.

To probe potential-dependent bulk oxidation mechanisms, we conducted operando X-ray absorption spectroscopy (Supplementary Fig. 16) at the Ir L$_3$-edge for Ir/TiO$_x$@Ti and unsupported Ir NPs under applied potentials (open circuit potential, OCP → 1.45 V). Figure 4f shows the X-ray absorption near-edge structure (XANES) white line intensity evolution, a direct indicator of Ir valence states. Both catalysts exhibit progressive white line enhancement and edge energy shifts with increasing potential, consistent with 5d-electron depletion (Ir$^0$ → Ir$^{4+}$ oxidation)[46]. However, Ir/TiO$_x$@Ti achieves near-complete oxidation with an average valence state of +3.22 at 1.45 V, approaching IrO$_2$'s +4 valence state (Fig. 4g and Supplementary Fig. 17). Its

oxidation is markedly faster than Ir NPs, confirming support-mediated bulk oxidation rather than surface-limited passivation.

Complementary extended X-ray absorption fine structure (EXAFS) analysis (Fig. 4g–i and Supplementary Table 2) resolves the local structural evolution under applied potentials, revealing distinct coordination dynamics between Ir/TiO$_x$@Ti and unsupported Ir NPs. For Ir/TiO$_x$@Ti, the Ir–O coordination number increases from 3.8 at OCP (0.8 V) to 5.6 at 1.45 V, approaching the ideal octahedral coordination (6) of rutile IrO$_2$. In contrast, unsupported Ir NPs exhibit a smaller Ir–O coordination increase (2.8 → 4.1) and retain higher Ir–Ir coordination at 1.45 V, indicative of surface-limited oxidation. These trends confirm that the TiO$_x$@Ti support facilitates deep oxygen penetration, driving full bulk oxidation of Ir nanoparticles rather than the conventional surface passivation observed in unsupported Ir NPs.

To resolve the bulk structural evolution of Ir/TiO$_x$@Ti during oxygen evolution, we performed time-resolved HAADF-STEM imaging across catalytic durations from 10 min to 100 h. Initial stages (OER-10 min, Fig. 5a) reveal Ir nanoparticles migrating into close proximity, contrasting their pristine dispersed state (Fig. 1d). Progressive coalescence forms an interconnected nanonetwork by 10 h (Supplementary Figs. 18–20), stabilizing into a fused architecture with single/twin boundaries after 100 h (Fig. 5b). This restructuring follows a two-stage mechanism (Fig. 5c): nanoparticle migration driven by interfacial energy minimization, followed by oriented attachment and lattice fusion[46–48]. Atomic-resolution HAADF-STEM (Fig. 5d and Supplementary Fig. 21) captures concomitant lattice expansion and distortion: the Ir(1$\bar{1}\bar{1}$) plane spacing increases from 2.26 Å (pristine) to 2.54 Å (OER-10 h), while interplanar angles α$_1$ and α$_2$ shift from 54.7° to 57.6° and 48.4°, respectively. These distortions align with operando EXAFS

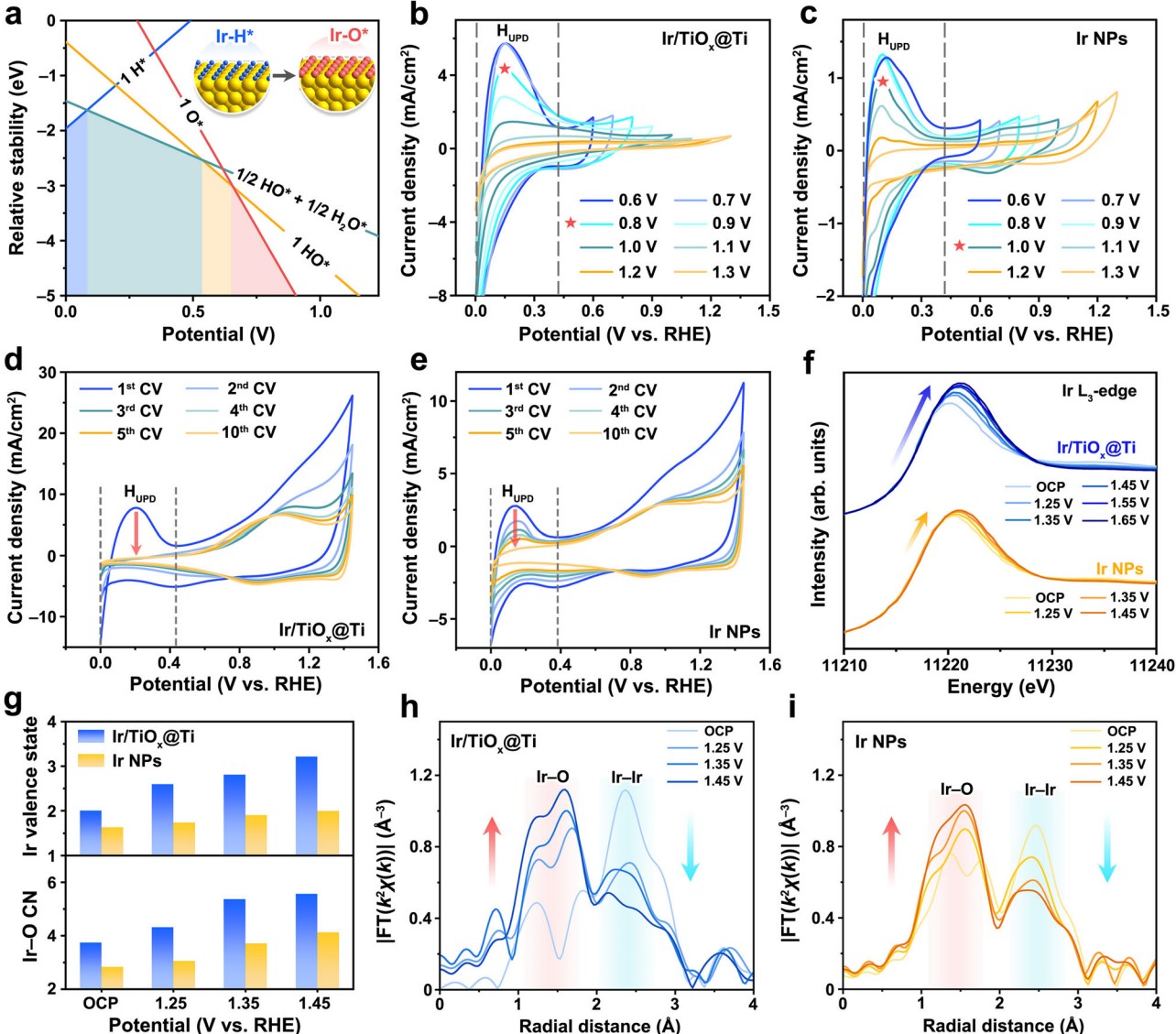

**Fig. 4 | Electrochemical oxidation behavior of supported and unsupported Ir nanoparticles. a** The surface phase diagram for Ir at pH 0. The ML means monolayer. The inset shows the structural model of 1 ML H* and 1 ML O* (abbreviated as 1 H* and 1 O*). **b, c** The CV curves of Ir/TiO$_x$@Ti and Ir NPs at the UPL of 0.6–1.3 V. 100 CV cycles are conducted for each potential window to achieve a relatively stable surface state. The CV curves of **d** Ir/TiO$_x$@Ti and **e** Ir NPs upon repetitive cycling at the UPL of 1.45 V. To eliminate the influence of Ir content, the Ir loading on the working electrode in (**b–e**) was fixed at 0.18 mg/cm$^2$. **f** Operando Ir L$_3$-edge XANES spectra of Ir/TiO$_x$@Ti and Ir NPs at varying potentials. **g** Ir valence state and coordination number (CN) of Ir/TiO$_x$@Ti and Ir NPs vary with the applied potential. The fitting methodology for the Ir valence state is described in the Methods section, and the reference spectra of Ir foil and IrO$_2$ are provided in Supplementary Fig. 17. Ir L$_3$-edge EXAFS spectra of **h** Ir/TiO$_x$@Ti and **i** Ir NPs in R space. The data are $k^2$-weighted and not phase-corrected.

evidence of oxygen intercalation into the Ir lattice, culminating in a bulk phase transition to rutile IrO$_2$ after 10 h. The final IrO$_2$(211) surface retains structural continuity with the precursor Ir(110) facet (Supplementary Fig. 22), rationalizing the preferential transformation pathway. The final crystallized IrO$_2$ structure on TiO$_x$@Ti leads to high stability during catalysis, exhibiting notably low Ir dissolution comparable to rutile IrO$_2$ (Supplementary Fig. 23).

While TiO$_x$@Ti facilitates bulk Ir crystallization, unsupported Ir NPs develop only a surface amorphous IrO$_x$ layer under identical conditions (Supplementary Fig. 24), consistent with prior reports[32,49]. To further investigate the significance of the unique TiO$_x$@Ti shell-core structure, we synthesized Ti$_4$O$_7$ (without a metallic Ti core) as a control material. Ir/Ti$_4$O$_7$ exhibit uniform and dense deposition of Ir NPs on the Ti$_4$O$_7$ surface. HAADF-STEM of the post-OER sample reveals that the Ir NPs on it, like unsupported Ir NPs, undergo surface amorphization (Supplementary Fig. 25). This contrast highlights the unique

role of the amorphous TiO$_x$ layer derived from metallic Ti in inducing phase transformation, a feature absent in crystalline Ti$_4$O$_7$-supported systems. These findings unequivocally demonstrate that the TiO$_x$@Ti support redirects Ir oxidation from surface passivation to bulk crystallization—a paradigm shift enabled by oxygen infusion into the nanoparticle core (Fig. 5e).

To elucidate the role of the TiO$_x$@Ti support in Ir oxidation, we employed in-situ $^{18}$O isotope labeling experiments to identify the oxygen source for Ir oxidation. The Ir/Ti$^{16}$O$_x$@Ti sample was initially employed to catalytic reaction in $^{18}$O-labeled H$_2$O for 10 h, yielding IrO$_2$/Ti$^{16}$O$_x$@Ti. In this process, the oxygen incorporated into the Ir lattice could originate from two potential pathways: (i) oxygen derived from water dissociation or (ii) oxygen migration from the oxide support. To determine the oxygen source in the formed IrO$_2$/Ti$^{16}$O$_x$@Ti, we conducted thermogravimetric coupled with mass spectrometry (TG-MS). During this analysis, IrO$_2$ decomposes into metallic Ir and O$_2$

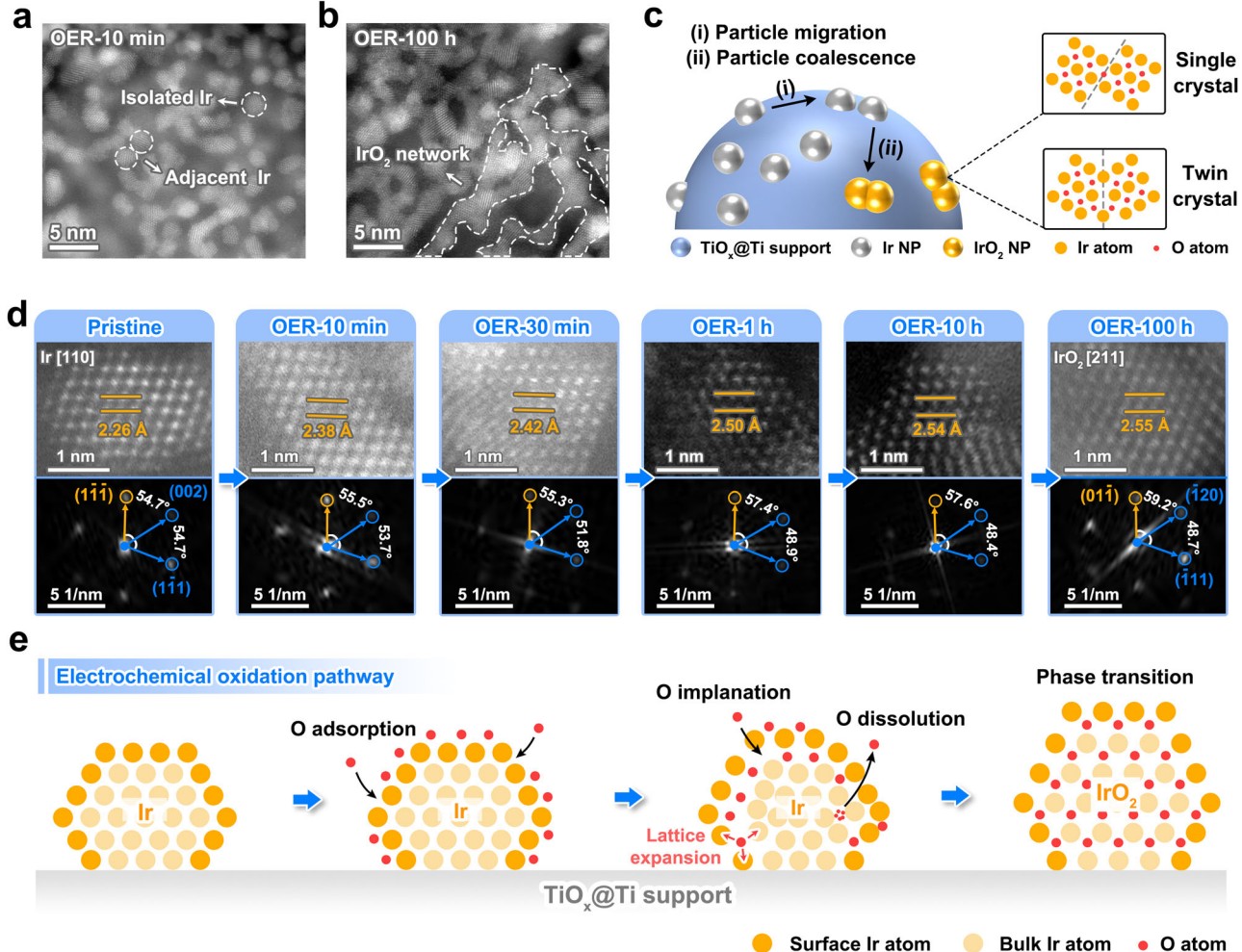

**Fig. 5 | Electrochemically induced structural evolution of Ir to rutile IrO₂ on TiOₓ@Ti.** The aberration-corrected HAADF-STEM images of **a** OER-10 min and **b** OER-100 h. **c** Schematic illustration of growth process of Ir nanoparticles on Ir/TiOₓ@Ti during oxidation. **d** High-resolution HAADF-STEM images and corresponding fast Fourier transform images of the six samples, including: Pristine, OER-10 min, OER-30 min, OER-1 h, OER-10 h, and OER-100 h. To strengthen the statistical robustness, three additional replicates of high-resolution and fast Fourier transform images are included for each post-catalysis sample in Supplementary Fig. 21. **e** Schematic illustration of structural changes during the transformation of metallic Ir to crystalline rutile IrO₂.

when heated to 1000 °C under an Ar atmosphere. The TG-MS results unambiguously showed that the evolved $O_2$ consisted exclusively of $^{36}O_2$ (Supplementary Fig. 26), confirming that the oxygen incorporated into the Ir lattice originated solely from water molecules. This finding, combined with CV and operando XAS results (Fig. 4), demonstrates that the TiOₓ@Ti support facilitates both oxygen covering on Ir surface and oxygen bulk diffusion in Ir lattice.

The atomic-scale mechanisms governing metal oxidation, particularly oxygen diffusion and lattice evolution, remain poorly understood. We propose that the unique shell-core architecture of TiOₓ@Ti might modulate the interfacial electronic structure to facilitate Ir oxidation. To probe these interfacial electronic effects, we conducted work function measurements, which revealed values of 4.8 eV for TiOₓ@Ti and 5.4 eV for Ir nanoparticles (Supplementary Fig. 27). This measurable difference in work functions establishes the thermodynamic driving force for electron transfer from the TiOₓ@Ti support to the supported Ir nanoparticles. Compared to unsupported Ir NPs, this electron transfer appears to weaken the originally strong Ir–O bonds by populating their antibonding orbitals[50,51], thereby could promote oxygen incorporation into the Ir lattice during OER. Furthermore, lower work function of metallic (4.3 eV)[52] establishes a sequential electron transfer pathway (Ti → TiOₓ → Ir), which may further enhance oxygen diffusion by reducing oxygen-binding strength.

These observations provide a possible electronic-structure-based explanation for the enhanced oxidation kinetics observed in Ir/TiOₓ@Ti.

## Catalytic mechanism for OER during reconstruction

To elucidate the OER mechanism during the electrochemical evolution of Ir on TiOₓ@Ti, we employed in-situ differential electrochemical mass spectrometry (DEMS) integrated with an $^{18}O$ isotope labeling strategy[53,54]. As depicted in Fig. 6a and Supplementary Fig. 28, the experimental protocol comprised three sequential phases. First, Ir/TiOₓ@Ti catalysts were electrochemically conditioned in unlabeled 0.1 M HClO₄ for durations spanning 0–10 h to generate samples with progressively oxidized Ir species. Subsequently, these pretreated catalysts were subjected to OER operation in an $^{18}O$-labeled electrolyte, enabling selective isotopic labeling of oxygen species associated with the oxidized Ir sites. Finally, the $^{18}O$-labeled catalysts were transferred to an unlabeled electrolyte for OER testing, where the oxygen isotopes in evolved $O_2$ were quantitatively monitored via DEMS. This approach allows mechanistic discrimination: if the reaction strictly adheres to AEM, lattice oxygen remains unlabeled, resulting exclusively in $^{32}O_2$ ($^{16}O^{16}O$) detection. Conversely, participation of LOM would permit $^{18}O$ incorporation into the catalyst lattice during the labeling phase, leading to the release of labeled

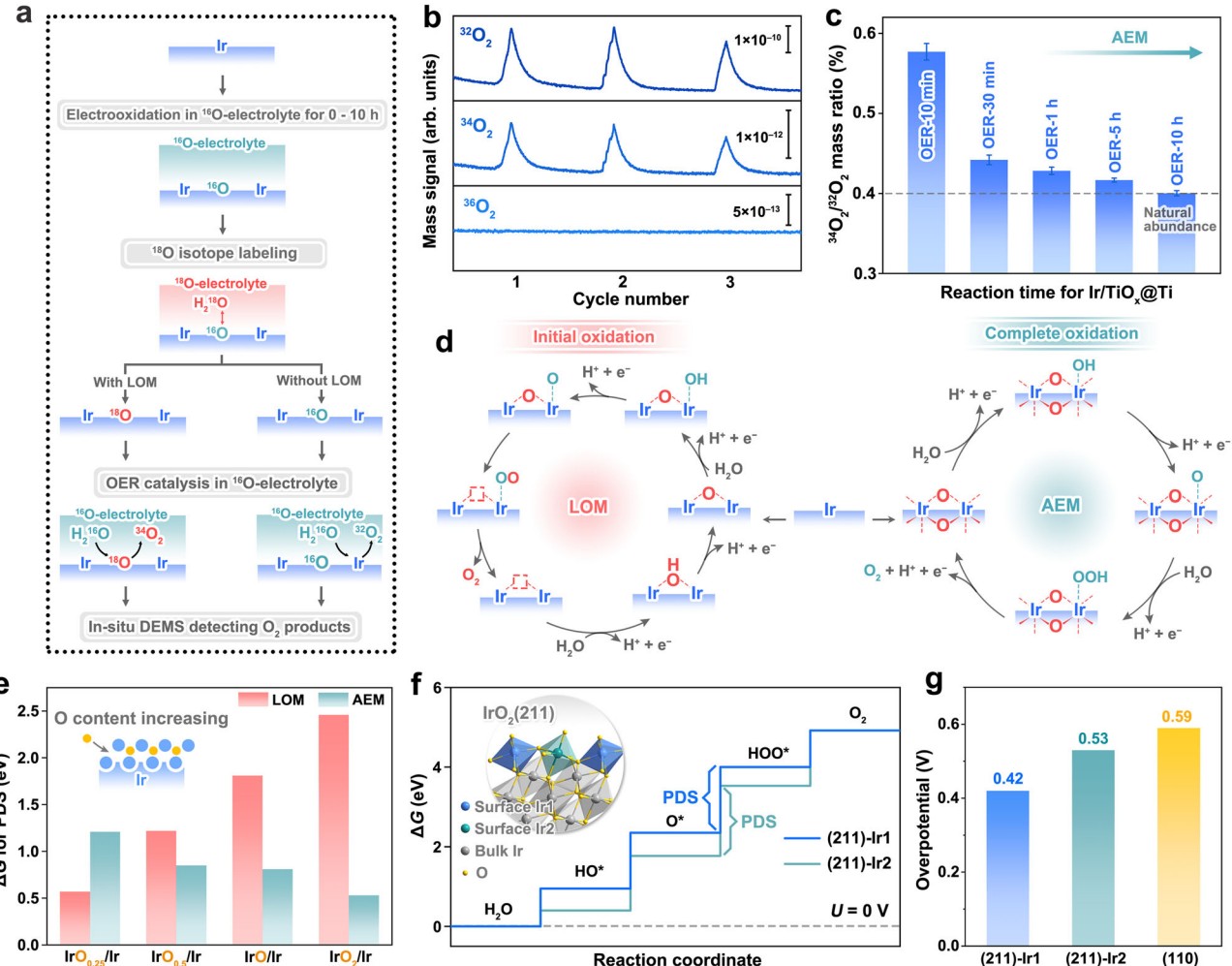

**Fig. 6 | Transformation of OER catalytic mechanism driven by Ir oxidation.**
**a** Schematic diagram of the process of in-situ DEMS combined with $^{18}$O isotope labeling experiments. **b** Mass signals of $^{32}O_2$, $^{34}O_2$ and $^{36}O_2$ for $^{18}$O-labeled Ir/TiO$_x$@Ti sample oxidized for 5 hours under OER conditions. **c** Mass ratios of $^{34}O_2$ to $^{32}O_2$ for Ir/TiO$_x$@Ti samples oxidized for different times under OER conditions. Error bars are drawn based on the standard deviations of three measurements. **d** Schematic illustration of LOM and AEM for oxidized Ir/TiO$_x$@Ti. **e** $\Delta G$ for the PDS of LOM and AEM pathways on O-implanted Ir with different oxygen content. The inset shows the schematic model of O-implanted Ir with varying surface oxygen content. **f** Gibbs free energy diagrams for OER of two Ir sites on IrO$_2$(211) surface at external bias $U = 0$ V. The inset displays the structural model of the IrO$_2$(211) surface, highlighting the two distinct active sites (Ir1 and Ir2). **g** The comparison of theoretical overpotentials for the (211) and (110) surfaces of IrO$_2$.

oxygen species during OER, as evidenced by the detection of $^{34}O_2$ ($^{18}O^{16}O$) and/or $^{36}O_2$ ($^{18}O^{18}O$).

DEMS analysis of Ir/TiO$_x$@Ti with varying oxidation degrees reveals distinct oxygen evolution pathways through $^{18}$O isotope tracing (Fig. 6b, c and Supplementary Fig. 29). The OER-10 min sample exhibits 0.58% $^{34}O_2$, which is above the 0.4% detection limit set by natural $^{18}$O abundance in water (0.2%)[55]. This confirms LOM participation. As oxidation progresses (10 min → 10 h), $^{34}O_2$ decreases to baseline levels (0.4%), suggesting a complete transition to AEM. This mechanistic shift correlates with structural stabilization: initial metastable oxygen in partially oxidized Ir lattices (OER-10 min, Fig. 6d, left) exhibits high mobility, favoring LOM-driven O$_2$ release. Progressive Ir oxidation enhances oxygen coordination stability, culminating in crystalline rutile IrO$_2$ (OER-10 h, Fig. 6d, right) where tightly bound lattice oxygen enforces strict AEM adherence. In contrast, unsupported Ir NPs retain persistent LOM activity (Supplementary Fig. 30) due to their amorphous IrO$_x$ shells' oxygen lability. The Ir/TiO$_x$@Ti's transition from the initial LOM-participated to the final complete AEM, which is absent in conventional catalysts, explains its enhanced stability, as AEM minimizes lattice oxygen loss and associated Ir dissolution during prolonged OER operation.

DFT calculations provide evidence consistent with the oxygen-content-dependent catalytic mechanism observed experimentally in Ir/TiO$_x$@Ti. As illustrated in Fig. 6e inset and Supplementary Fig. 31, we modeled near-surface Ir lattices with incremental oxygen stoichiometries (IrO$_{0.25}$/Ir → IrO$_2$/Ir) to simulate progressive oxidation. Gibbs free energy ($\Delta G$) analysis of the potential-determining steps (PDS) reveals a mechanistic crossover: the LOM barrier increases monotonically from 0.6 eV (IrO$_{0.25}$) to 2.5 eV (IrO$_2$), while the AEM barrier decreases from 1.2 eV to 0.5 eV (Fig. 6e, Supplementary Table 3 and Supplementary Data 1). This establishes a mechanistic shift: low oxygen content (<IrO$_{0.5}$) favors LOM through labile lattice oxygen participation, whereas progressive oxidation stabilizes Ir−O coordination, suppressing the activity of lattice oxygen and thereby enforcing AEM via surface-bound intermediates. These theoretical predictions align precisely with in-situ DEMS data, suggesting that the suppression of LOM during Ir/TiO$_x$@Ti restructuring arises from oxygen-induced electronic and structural stabilization−a synergy absent in conventional catalysts with static amorphous phases.

Following the mechanistic and structural elucidation of Ir/TiO$_x$@Ti, we computationally evaluated the OER activity of its final stable phase−rutile IrO$_2$(211)−using the four-step reaction

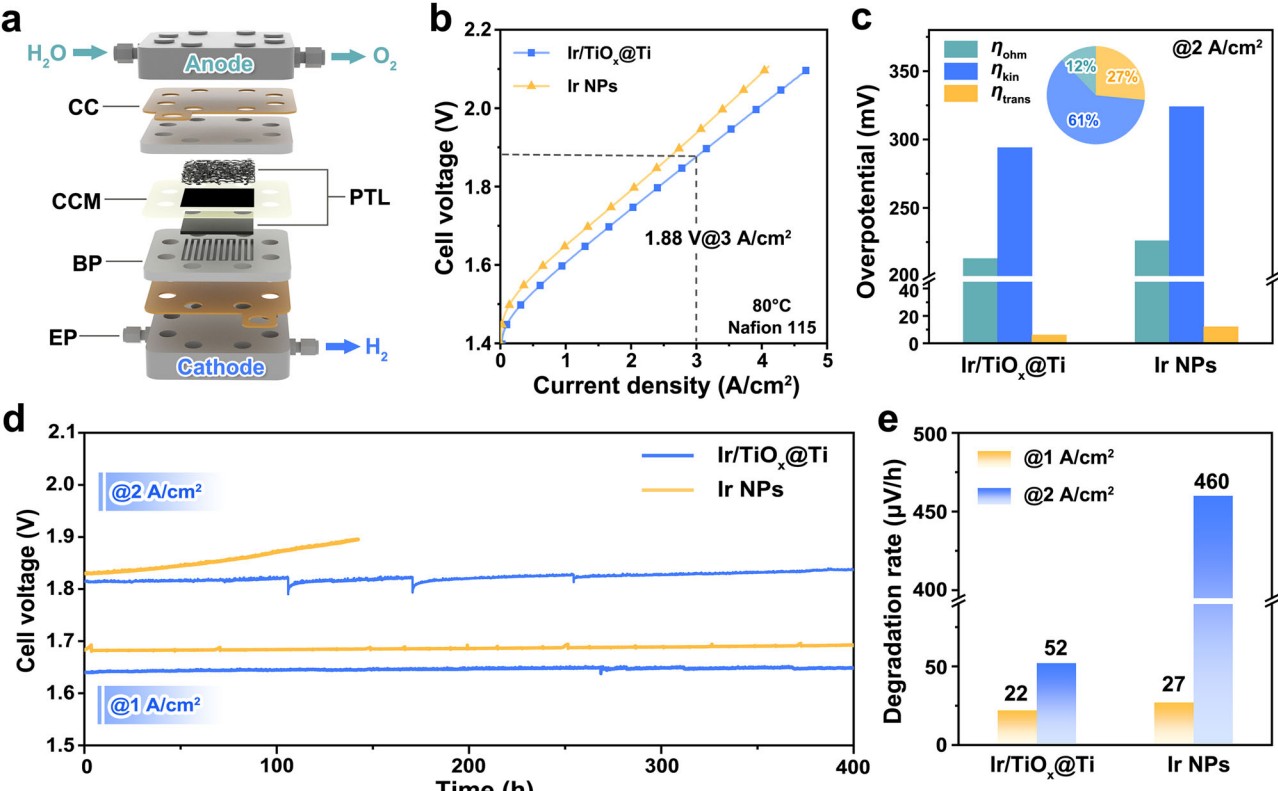

**Fig. 7 | OER performance in PEMWE devices. a** Schematic diagram of PEMWE. CC, BP, PTL and EP represent current collector, bipolar plate, porous transport layer, and end plate, respectively. **b** Steady-state polarization curves of PEMWE using Ir/TiO$_x$@Ti and Ir NPs as anode catalysts. **c** Comparison of ohmic overpotential ($\eta_{ohm}$), kinetic overpotential ($\eta_{kin}$), and transport overpotential ($\eta_{trans}$) of PEMWE using Ir/TiO$_x$@Ti and Ir NPs anodes. The inset illustrates the percentage contributions of the three decoupled overpotentials to the total overpotential reduction observed in the Ir/TiO$_x$@Ti-based PEMWE compared to the Ir NPs-based PEMWE. **d** Chronopotentiometry curves of PEMWEs using Ir/TiO$_x$@Ti and Ir NPs anode operated at 1 and 2 A/cm$^2$ current densities. **e** Voltage degradation rates of PEMWEs using Ir/TiO$_x$@Ti and Ir NPs anodes.

framework[56,57]. For benchmarking, the thermodynamically favored IrO$_2$(110) surface was analyzed in parallel. The IrO$_2$(211) facet hosts two distinct Ir active sites, both governed by O–O bond formation (O* + H$_2$O → HOO* + H$^+$ + e$^-$) as the potential-determining step (Fig. 6f and Supplementary Data 1). Theoretical overpotentials for these sites are calculated as 0.42 V and 0.53 V, lower than the 0.59 V overpotential of IrO$_2$(110) (Fig. 6g, Supplementary Fig. 32 and Supplementary Data 1). This reduction in overpotential arises from the (211) facet's unique atomic configuration: its stepped geometry exposes under-coordinated Ir sites that optimize HOO* adsorption energetics, while the conventional (110) facet's flat terrace imposes stronger inter-mediate binding. These findings rationalize the experimentally observed high OER activity of Ir/TiO$_x$@Ti, implying how support-driven restructuring generates non-traditional active facets inaccessible to bulk-synthesized catalysts.

**PEMWE device performance**

Leveraging the enhanced OER activity and stability of Ir/TiO$_x$@Ti, we fabricated proton exchange membrane water electrolyzers (PEMWEs) with an Ir loading of 0.3 mg$_{Ir}$/cm$^2$ via catalyst-coated membrane (CCM) assembly (Fig. 7a). The anode (Ir/TiO$_x$@Ti) and cathode (Pt/C) layers, uniformly deposited on Nafion 115 membranes (Supplementary Figs. 33–36), form porous agglomerates with a controlled thickness of ~2.5 μm, as verified by cross-sectional SEM and X-ray fluorescence spectroscopy (XRF) quantification. At 80 °C, the Ir/TiO$_x$@Ti-based cell achieves a current density of 4.0 A/cm$^2$ at 2.0 V, outperforming unsupported Ir NPs (3.5 A/cm$^2$@2.0 V, Fig. 7b) and delivering a high Ir-specific power of ≈14 kW/g$_{Ir}$ at 70% LHV efficiency[58] (1.79 V). This performance benchmark places Ir/TiO$_x$@Ti ahead of the most recent

reports of supported Ir catalysts, such as Ir/TiO$_2$, Ir/Sb-SnO$_2$ and Ir/Nb$_2$O$_{5-x}$ (Supplementary Table 4)[23,25,42,59–66].

Electrochemical impedance spectroscopy (EIS) deconvolution reveals the origin of Ir/TiO$_x$@Ti's higher PEMWE performance, distin-guishing contributions from ohmic resistance, mass transport, and kinetic overpotentials (Supplementary Figs. 37 and 38)[67–69]. As shown in Fig. 7c, the Ir/TiO$_x$@Ti anode exhibits a 30 mV reduction in kinetic overpotential compared to Ir NPs (294 versus 324 mV at 2 A/cm$^2$), contributing to 61% of the total overpotential mitigation, directly attributable to its high intrinsic activity and high Ir utilization effi-ciency. Concurrently, the mass transport overpotential is reduced by 27% (6 mV versus 12 mV), enabled by the hierarchical porosity of Ir/TiO$_x$@Ti agglomerates—a 2.5 μm-thick catalyst layer with inter-connected pores facilitating rapid bubble detachment and proton diffusion. These dual enhancements collectively enable the Ir/TiO$_x$@Ti-based cell to achieve industry-leading current densities (4.0 A/cm$^2$@2.0 V).

The operational stability of Ir/TiO$_x$@Ti-based PEMWEs was evaluated via chronopotentiometry at industrially relevant current densities (1–2 A/cm$^2$, Fig. 7d). The cell exhibits very low voltage degradation over 400 h at both 1 A/cm$^2$ (degradation rate: 22 μV/h, Figs. 7e) and 2 A/cm$^2$ (degradation rate: 52 μV/h), achieving an 89% reduction in degradation rate compared to unsupported Ir NPs (460 μV/h). Additionally, commercial Ir black and IrO$_2$/TiO$_2$ cata-lysts were tested. Under low Ir loadings, Ir/TiO$_x$@Ti still demon-strates markedly higher activity and stability advantages compared to both commercial Ir black and IrO$_2$/TiO$_2$ (Supplementary Figs. 39 and 40). The combined metrics, including low Ir loading, high current density, and good catalytic durability, establish

Ir/TiO$_x$@Ti as a promising anode material for green hydrogen production.

## Discussion

In summary, our integrated operando study, combining spectroscopic, electrochemical, and computational approaches, deciphers the structural evolution of TiO$_x$@Ti-supported Ir nanoparticles during acidic oxygen evolution. For nearly half a century, the field has operated under the assumption that surface amorphization into hydrous IrO$_x$ is an unavoidable consequence of Ir-catalyzed OER. Our results overturn this paradigm by revealing a support-directed bulk crystallization pathway: metallic Ir undergoes a rapid, full-phase transition to (211)-oriented rutile IrO$_2$, concomitant with a self-adaptive mechanism shift from lattice oxygen participation (LOM) to adsorbate-mediated pathways (AEM). This paradigm shift not only resolves the long-standing activity-stability conflict but also repositions catalyst supports as dynamic architects of active phases.

## Methods

### Chemicals and materials

Ethylene glycol (C$_2$H$_6$O$_2$, AR) was purchased from Beijing Chemical Factory. Absolute ethanol (C$_2$H$_6$O, AR) and isopropyl alcohol (C$_3$H$_8$O, AR) were purchased from Sinopharm Chemical Reagent Co., Ltd. Ti nanospheres (Ti, ≥99.8% metals basis, 60 nm), nano titanium dioxide (TiO$_2$, anatase, 5–10 nm, ≥99.8% metals basis) and lithium chloride (LiCl, AR, ≥99.0%) were obtained from Shanghai Aladdin Biochemical Technology Co., Ltd. Hexachloroiridium acid solution (H$_2$IrCl$_6$·xH$_2$O, Ir ≥35 wt%) was purchased from Shanghai Yurui Chemical Co., Ltd. Perchloric acid (HClO$_4$, 70.0–72.0%) was purchased from Tianjin Xinyuan Chemical Co., Ltd, and Pt/C (40 wt%) catalyst was supplied by Johnson Matthey Company. Nafion® perfluorinated resin solution (D520, 5 wt%) was purchased from Sigma-Aldrich. The glassy carbon electrode (CHI104), platinum wire electrode (CHI115), and saturated calomel electrode (CHI150) were purchased from Shanghai CH Instruments Co., Ltd., while the platinum clip electrode was purchased from Tianjin Ida Hengsheng Technology Development Co., Ltd. The waterproof and breathable membrane (PTFE material, pore size ≤20 nm, porosity ≥50%) was purchased from Linglu Instruments (Shanghai) Co. Ltd. All the chemicals and reagents were used without further purification. Highly purified water (resistivity >18 MΩ·cm) was sourced from a PALL PURELAB Plus system.

### Materials synthesis

Ir/TiO$_x$@Ti was synthesized using the polyol reduction method. First, 200.0 mg of Ti nanospheres was added to 30 mL of ethylene glycol. After ultrasonic dispersion for 1 h, the suspension was transferred to a 250 mL three-neck flask and stirred continuously at 180 °C for 1 h. A solution of 244.9 mg of H$_2$IrCl$_6$·xH$_2$O in 4 mL of ethylene glycol was then injected into the flask, and the mixture was stirred for 3 hours at the same temperature. Once the mixture naturally cools to room temperature, it is centrifuged to isolate the product, which is then washed five times with water to remove any residual organic compounds. The product was then dried in an oven at 80 °C for 4 h, yielding a sample with 24 wt% Ir content. To achieve Ir contents of 9 wt%, 16 wt%, and 31 wt%, the amounts of H$_2$IrCl$_6$·xH$_2$O used were 63.5 mg, 142.9 mg, and 381.0 mg, respectively. In addition, unsupported Ir NPs were synthesized following a similar procedure, except without the inclusion of Ti nanospheres. TiO$_x$@Ti was synthesized in the same manner, but without the addition of H$_2$IrCl$_6$·xH$_2$O.

The synthesis of Ti$_4$O$_7$ involved the following steps: 50.0 mg of Ti nanospheres, 584.0 mg of nano TiO$_2$, and 634.0 mg of LiCl were thoroughly ground in mortar to ensure homogeneity. The mixed powder calcined at 600 °C for 2 h under an argon atmosphere, followed by natural cooling to room temperature. The resulting product was washed three times with deionized water, filtered, and dried to obtain blue-black Ti$_4$O$_7$ powder. Ir/Ti$_4$O$_7$ was synthesized under identical conditions to those described above for Ir/TiO$_x$@Ti.

### Characterizations

XRD patterns were collected using a Rigaku D/Max 2550/PC X-ray diffractometer with Cu Kα radiation ($\lambda$ = 1.5418 Å). SEM images were captured using a field emission SEM (JEOL 7800F), and EDX analysis was carried out on the connected system. TEM images and element mapping were obtained on a JEM-2100F, equipped with a field emission gun operating at 200 kV. Aberration-corrected HAADF-STEM images acquired using JEM-ARM300F operating at 300 kV. Raman spectra were obtained on a Renishaw Raman system model 1000 spectrometer, utilizing a 20 mW air-cooled argon-ion laser with a 532 nm wavelength for excitation. ICP-OES measurements were conducted on a PerkinElmer Optima 3300DV ICP spectrometer. TG-MS measurements were conducted on NETZSCH STA449 F3+Quadrupole Mass Spectrometry (QMS403D) in Ar. The work function was determined using a scanning Kelvin probe system (KP Technology Ltd.) in air. XPS measurements were performed on a Thermo Fisher Scientific ESCALAB 250Xi photoelectron spectroscopy system equipped with a monochromatic Al Kα (1486.6 eV) X-ray source. In-situ DEMS measurements were executed on an in-situ differential electrochemical mass spectrometer from Linglu Instruments (Shanghai) Co. Ltd.

Quasi-in-situ Raman spectroscopy is conducted in which Ti nanospheres and ethylene glycol (EG) are heated in air at 180 °C, and a certain amount of suspension is withdrawn from the reaction system at various time intervals (ranging from 1 to 90 min), with the powder product, after cleaning, analyzed using Raman spectrometer.

PDF measurements were carried out by packing samples into 0.7 mm borosilicate capillaries, with diffraction data acquired using a Mythen-II detector over 115°. The wavelength, zero error, and instrument contribution to the peak profile were established using the NIST Si sample with a refined wavelength of 0.5903 Å. Background subtraction during the PDF experiments was performed using an empty capillary.

Operando XANES and EXAFS measurements of Ir L$_3$-edge were performed in the fluorescence mode at the beamline of 1W1B in Beijing Synchrotron Radiation Facility (BSRF). Data reduction, analysis, and EXAFS fitting were carried out using the Athena and Artemis software packages. For EXAFS modeling, R-space (1.0–3.5 Å) and Fourier transforms in k-space (2.0–12.0 Å$^{-1}$) were applied, with the amplitude reduction factor S$_0^2$ set at 0.815 to determine the coordination numbers (CNs) for the Ir–O/Ir scattering path in the samples. The data were fitted in R-space with theoretical models based on the crystal structure of rutile IrO$_2$ and metallic Ir. The valence state fitting for XANES was performed by establishing a linear relationship between the white-line peak intensity of the Ir L$_3$-edge and reference standards (metallic Ir and IrO$_2$)[70].

The identification of the (211) exposed facet was achieved through zone axis indexing methodology, which integrates high-resolution lattice imaging, FFT analysis, and crystallographic zone law. The procedural workflow comprises three key phases: (1) Cross-lattice fringe regions were identified in HRTEM images by locating intersections of three distinct lattice fringes, with their spacings and interplanar angles measured. (2) FFT analysis was applied to these regions to precisely map diffraction spots, which were then compared to theoretical lattice projections. (3) The zone axis (exposed facet) was determined using the zone law, calculated by cross-multiplying the Miller indices of adjacent lattice planes.

To clarify the oxygen source during Ir oxidation, we designed an experimental protocol combining $^{18}$O isotope labeling with thermogravimetric-mass spectrometry (TG-MS). The principle involves subjecting Ir/TiO$_x$@Ti to OER catalysis in an $^{18}$O-labeled electrolyte for 10 h to form $^{18}$O-labeled Ir/TiO$_x$@Ti. Subsequent TG was performed to thermally decompose the sample, releasing oxygen from

$IrO_2$ as $O_2$ gas at high temperatures. The evolved $O_2$ was then analyzed by MS to determine its mass-to-charge (m/z) ratios. Potential $O_2$ isotopic signatures include $^{32}O_2$ ($^{16}O^{16}O$), $^{34}O_2$ ($^{16}O^{18}O$), and $^{36}O_2$ ($^{18}O^{18}O$). Detection of $^{32}O_2$ or $^{34}O_2$ would indicate oxygen migration from the support ($^{16}O$) into the Ir lattice, whereas exclusive detection of $^{36}O_2$ would confirm water as the sole oxygen source. The experimental procedure is as follows: First, the $Ir/TiO_x@Ti$ catalyst was subjected to OER in 0.1 M $HClO_4$ prepared with $H_2^{18}O$ for 10 h. The sample was then washed, centrifuged, and dried to obtain $^{18}O$-labeled $Ir/TiO_x@Ti$. A controlled mass of the labeled sample was analyzed using TG-MS. For comparison, pristine (unlabeled) $Ir/TiO_x@Ti$ was also tested.

### Electrochemical measurements in a three-electrode configuration

Catalyst performance was evaluated in a standard three-electrode system using a CH Instrument (Model 760E). Raw data were directly recorded by the instrument's proprietary software and subsequently underwent manual unit conversion. Where error bars are shown, the data represent the results of at least three independent measurements; data points without error bars represent single measurements.

The electrolyte consisted of 0.1 M $HClO_4$ solution bubbled with $O_2$ gas. The volume of the single-compartment electrochemical cell was fixed at 30 mL. The 0.1 M $HClO_4$ electrolyte was prepared by first diluting 40.8 mL of concentrated $HClO_4$ to 500 mL with ultrapure water to obtain a 1 M $HClO_4$ solution, followed by diluting 50 mL of this intermediate solution to 500 mL with ultrapure water. Freshly prepared electrolyte was immediately validated for pH using a calibrated FiveEasy Plus FE28 pH meter (accuracy ± 0.01), yielding three measurements (pH 1.03, 1.00, and 0.99) with a mean of 1.01 ± 0.02 (mean ± standard deviation), confirming compliance with the target pH ~1.00. The electrolyte was stored in glass bottles in a dark, cool environment at ambient temperature. Prior to each use, pH stability was verified; deviations exceeding ±0.05 from the initial mean pH necessitated solution replacement to ensure consistency.

Linear sweep voltammetry (LSV) measurements were conducted at a scan rate of 1 mV/s with 85% iR-drop compensation applied to correct for all electrolyte/contact resistance of the setup[71,72], where the compensation values were directly calculated and output by the electrochemical workstation to automatically correct the LSV curves, rather than being manually adjusted through EIS impedance testing. Specifically, for different samples during LSV testing, the actual applied compensation values were as follows: for the 9% Ir sample, three measured compensation values were 39.7, 38.4, and 40.7 Ω with an average of 39.7 ± 1.2 Ω; for the 16% Ir sample, values were 32.7, 33.1, and 33.0 Ω averaging 32.9 ± 0.2 Ω; for the 24% Ir sample, values were 32.6, 32.8, and 33.8 Ω averaging 33.1 ± 0.6 Ω; for the 31% Ir sample, values were 33.1, 33.2, and 33.1 Ω averaging 33.1 ± 0.1 Ω; and for the Ir NPs sample, values were 35.7, 34.8, and 34.9 Ω with an average of 35.1 ± 0.5 Ω.

Chronopotentiometry curves were recorded without iR-drop compensation. A glassy carbon electrode (GCE) served as the working electrode (WE), a Pt wire as the counter electrode (CE), and a saturated calomel electrode (SCE) as the reference electrode (RE). All potentials were referenced to the reversible hydrogen electrode (RHE), as shown in Eq. (1):

$$E_{vs.RHE} = E_{vs.SCE} + 0.246V \qquad (1)$$

where 0.246 V vs. SCE represents the potential of zero net current. The SCE was calibrated according to the method established by Boettcher et al.[73], wherein a three-electrode system was employed using the operational electrolyte: a Pt working electrode, Pt counter electrode, and the SCE as the reference electrode. Hydrogen gas was continuously bubbled through the working electrode for a minimum of 30 min to establish a stable hydrogen environment. Subsequently, the

zero-current potential was determined through a combination of slow-scan voltammetry (10 mV/s) and constant-potential steps. This measured potential corresponds to the calibrated SCE potential relative to the RHE, yielding a value of 0.246 V in this work.

For the 9% Ir, 16% Ir, 24% Ir, 31% Ir, and Ir NPs samples, 19.0 mg, 10.7 mg, 7.1 mg, 5.5 mg, and 1.8 mg of catalysts were weighed and dispersed in 400 μL isopropyl alcohol containing 10 μL of a 5 wt% Nafion solution, respectively. Then, 3 μL of catalyst ink was applied to the surface of the GCE, with a catalytic area of 0.071 cm², and allowed to dry naturally. The Ir loading on the WE was 0.18 mg/cm². For example, the calculation for the 9% Ir sample proceeded as follows: 19.0 mg × 9% × 3 μL/400 μL/0.071 cm² = 0.18 mg/cm². To prevent GCE passivation, a piece of carbon paper with an active area of 0.09 cm², loaded with catalysts, was used as the WE for the chronopotentiometry test, with a Ir loading of 0.18 mg/cm². For accurate evaluation of Ir leaching during the OER process, the catalyst ink was dropped onto a carbon paper with an active area of 1 cm² and a catalyst loading of 10 mg/cm². The WE was then catalyzed for 10 h at 1.55 V vs. RHE, and the electrolyte was collected at the first hour and every 2 h for ICP-OES analysis. The scan rate for all CV tests was set to 50 mV/s.

To calculate $j_{geo}$ of the catalysts, the measured current was normalized by the geometric area of the GCE, as described in Eq. (2):

$$j_{geo} = \frac{i}{s}(mA/cm_{geo}^2) \qquad (2)$$

where $i$ represents the measured current and $s$ is the geometric area of the GCE.

To calculate the $j_{Ir}$ of the catalysts, the measured current was normalized by the mass of Ir on the GCE, as described in Eq. (3):

$$j_{Ir} = \frac{i}{m \times Ir(wt\%)}(mA/g_{Ir}) \qquad (3)$$

where $m$ represents the catalyst loading mass on the GCE, and $Ir$ (wt%) represents the mass fraction of Ir in the catalyst.

$Ir/TiO_x@Ti$ samples with different oxidation times were prepared for HAADF-STEM on GCE to ensure uniform oxidation. Specifically, the Ir loading on a GCE with an active area of 0.071 cm² was controlled at 0.18 mg/cm², and the electrode was oxidized in 0.1 M $HClO_4$ at 1.55 V for varying durations. Finally, the catalyst on the GCE was ultrasonically dispersed in an ethanol solution for collection.

In-situ DEMS testing was carried out in a three-electrode cell using 0.1 M $HClO_4$ solution as the electrolyte[53,54]. (i) The pristine $Ir/TiO_x@Ti$ sample was catalyzed on a glassy carbon electrode in unlabeled 0.1 M $HClO_4$ solution at 1.55 V for varying durations, specifically 0 min, 20 min, 50 min, 4 h 50 min, and 9 h 50 min; (ii) The oxidized $Ir/TiO_x@Ti$ was labeled with the $^{18}O$ isotope in 0.1 M $HClO_4$ solution using $H_2^{18}O$ as the solvent at 1.55 V for 10 minutes. Therefore, the total oxidation time of the catalyst is the sum of the durations from steps (i) and (ii), resulting in oxidation times of 10 min, 30 min, 1 h, 5 h, and 10 h, respectively; (iii) The labeled electrode was washed with $H_2^{16}O$ to remove residual $H_2^{18}O$ physically adhered to the catalyst surface. Afterward, CV measurements were conducted at a scan rate of 50 mV/s within the potential range of 0.6 V to 1.2 V vs. RHE to remove the adsorbed $^{18}O$ from the catalyst surface. (iv) For the labeled electrode, three LSV cycles were performed in unlabeled 0.1 M $HClO_4$ solution, within the potential range of 1.2–1.7 V vs. RHE. The mass spectrometer was used to monitor the gaseous products, including $^{36}O_2$, $^{34}O_2$, and $^{32}O_2$. The fractions of these products were calculated based on the integrated areas of the corresponding mass signals.

### Electrochemical measurement of PEMWE

Nafion 115 membrane was used to fabricate the catalyst-coated membrane (CCM). The pre-treatment process involved treating the N115

membrane with 3 wt% $H_2O_2$, highly purified water, and 0.5 M $H_2SO_4$ at 80 °C for 1 hour each. Following treatment, the membrane was rinsed with highly purified water. $Ir/TiO_x@Ti$, Ir NPs, commercial Ir black and $IrO_2/TiO_2$ were used as the anode catalysts, while commercial 40 wt% Pt/C was used as the cathode catalyst. The catalysts were ultrasonically dispersed in 10 mL deionized water and 15 mL isopropyl alcohol. Nafion ionomer was then added separately to achieve mass fractions of 12 wt% in the anode catalyst ink and 35 wt% in the cathode catalyst ink. The catalyst inks were applied to both sides of the N115 membrane using an ultrasonic spraying instrument. The CCM was then hot-pressed at 130 °C and 10 MPa for 3 minutes to fabricate the membrane-electrode assembly (MEA) for testing. X-ray fluorescence (XRF) spectroscopy confirmed that the mass loading of the anode precious metal Ir and the cathode precious metal Pt were both controlled at 0.3 mg/cm². The PEMWE cell fixture was sourced from Hefei Momenta Energy Co., Ltd. The MEA was assembled in a single cell with an accessible area of 5 cm². For the anode, a 190 μm porous Ti plate coated with Pt (from Hefei Momenta Energy Co., Ltd.) was employed as the porous transport layer (PTL), while a 180 μm carbon paper served as the PTL for the cathode.

The PEMWE was operated at 80 °C with deionized water circulating through a peristaltic pump at rate of 60 mL/min. During the test, the MEA was activated by electrolysis at constant currents of 0.1 A/cm² for 1 h, 0.5 A/cm² for 1 h, and at a fixed potential of 1.70 V for 1 h. Steady-state polarization curve was recorded using a Gamry instrument at a scan rate of 20 mV/s. To assess the long-term catalytic stability of electrolytic cells based on $Ir/TiO_x@Ti$ or Ir NPs catalysts, chronopotentiometry was performed using the NEWARE battery test system (CT-4008-5V100A, Shenzhen, China) at constant currents of 1 A/cm² and 2 A/cm². For SEM imaging of the MEA cross-section, a 2 × 2 cm sheet of the MEA was cut and frozen in liquid nitrogen for ~1 min. The catalyst-coated area was then separated using tweezers. Once the sample was completely dry, the cross-section was mounted on the specimen stage for imaging.

The voltage loss of a PEMWE, also referred to as overpotential, comprises three components: ohmic loss ($\eta_{ohm}$), kinetic loss ($\eta_{kin}$), and mass transfer loss ($\eta_{trans}$)[67–69]. Ohmic loss is represented by the overall ohmic resistance of the PEMWE device, as shown in the following Eq. (4):

$$\eta_{ohm} = j \times HFR (V) \qquad (4)$$

where $j$ represents current density, and HFR denotes high-frequency resistance, derived from electrochemical impedance spectroscopy (EIS). To calculate $\eta_{kin}$, the following steps are taken. First, the voltage from the PEMWE's polarization curve is corrected by subtracting $\eta_{ohm}$ to obtain $E_{ohmic-corrected}$. Next, the Tafel slope b is determined by fitting the $E_{ohmic-corrected}$-log(j) curve at low current densities (<100 mA/cm²), from which the exchange current density $j_0$ is extracted. The tafel curve was extrapolated to the maximum current density to determine $E_{kin}$. This analysis assumes that HER is non-polarizable, so the total $\eta_{kin}$ is attributed solely to the OER. Finally, the $\eta_{kin}$ is calculated using the following Eq. (5):

$$\eta_{kin} = b \times \log\left(\frac{j}{j_0}\right) (V) \qquad (5)$$

where $j$ denotes the applied current density.

The $\eta_{trans}$ is calculated by the following Eq. (6):

$$\eta_{trans} = E_{ohmic-corrected} - E_{kin} (V) \qquad (6)$$

## Computation details
All DFT calculations were performed using the Vienna Ab initio Simulation Package (VASP 5.4.4)[74,75]. The electron-ion interactions were described using the projector augmented wave (PAW) method[76]. The generalized gradient approximation (GGA) with the Perdew-Burke-Ernzerhof (PBE) functional was applied[77]. For the metals Ti and Ir, the Ti_sv and standard Ir pseudopotentials were employed. For all calculations, the cutoff energy of plane-wave basis was set to 400 eV, with convergence criteria of $10^{-4}$ eV for energy and 0.05 eV/Å for force. The Brillouin zones were sampled by Monkhorst–Pack scheme[78] with a k-point separation of $0.03 \times 2\pi$ Å⁻¹. A 15 Å thick vacuum layer was added between the slabs in all slab model calculations to prevent interlayer interactions. Symmetrization was disabled, and a dipole correction was applied. The DFT-D3 method with the Grimme scheme was used to account for van der Waals (vdW) interactions[79].

For the metallic Ti slab model, a $4 \times 4 \times 2$ supercell was used for the Ti (001) surface; for the metallic Ir slab model, a $2 \times 1 \times 4$ supercell was used for the Ir (110) surface; and for the rutile $IrO_2$ slab model, a $1 \times 1 \times 6$ supercell was used for the $IrO_2$ (211) plane. A $1 \times 2 \times 4$ supercell was used for the $IrO_2$(110) plane, with half of the atomic layers kept fixed.

The construction of Ir slab models with varying oxygen contents is described as follows: The first atomic layer of each model consists of four Ir atoms, which form the computational basis for composition calculations. To simulate surface coverage under oxidizing conditions, four adsorbed oxygen atoms were placed on the topmost layer of each model; these adsorbed oxygen atoms were excluded from the compositional analysis. Oxygen atoms were then incrementally introduced into the subsurface layer (1, 2, 4, or 8 oxygen atoms) to achieve the desired stoichiometries: $Ir_4O$ ($IrO_{0.25}$), $Ir_4O_2$ ($IrO_{0.5}$), $Ir_4O_4$ (IrO), and $Ir_4O_8$ ($IrO_2$). For each oxygen content, multiple configurations were generated by randomly inserting subsurface oxygen atoms, and the lowest-energy configuration was selected for further analysis.

Adsorption energy change of oxygen ($\Delta E_O$) is defined as the total energy difference between the optimized configurations before and after oxygen migration, which can be termed the oxygen adsorption energy difference or oxygen binding energy difference.

For all calculations related to the Gibbs free energy change of reactions, including the phase diagram on the Ir surface, the AEM and LOM pathways for oxidized Ir, and the theoretical activity of AEM on the (211) and (110) faces of rutile $IrO_2$, the following Eq. (7) was applied:

$$\Delta G = \Delta E + \Delta ZPE + T\Delta S - eU \qquad (7)$$

where $\Delta E$ represents the total energy change of the reaction calculated using DFT, $\Delta ZPE$ and $\Delta S$ are the differences in zero-point energy and entropy of the reaction, respectively, and T is the absolute temperature. The effect of the external bias $U$ on the reaction is incorporated as $-eU$ term[56,57,80].

## Data availability
The data that support the findings of this study are available within the article and its Supplementary Information. All other relevant data supporting the findings of this study are available from the corresponding authors upon request. Source data are provided with this paper.

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

## Acknowledgements

This work was supported by the National Key R&D Program of China (no. 2021YFB4000200), the National Natural Science Foundation of China (NSFC) (no. 22179046 and 22279040), the Jilin Province Science and Technology Development Plan (no. 20220203086SF), and the Fundamental Research Funds for the Central Universities. We thank the 1W1B-XAFS Beamline of Beijing Synchrotron Radiation Facility (https://cstr.cn/31109.02.BSRF.1W1B) for providing technical support and assistance in XAFS data collection.

## Author contributions

X.X.Z. and H.C. conceived the project and designed the experiments. K.Z. carried out the chemical synthesis and electrochemical experiments, and performed the theoretical calculations. X.L. helped with experimental design, in-situ experimental investigations, and figure design. Y.W. performed the operando XANES and EXAFS measurements. Y.Z. and X.Z. assisted with structural analysis and provided valuable suggestions for the research. K.Z., H.C. and X.X.Z. wrote the paper. All authors discussed the results, drew conclusions and commented on the manuscript.

## Competing interests

The authors declare no competing interests.

## Additional information

Hui Chen or Xiaoxin Zou.

