## [Transparent Peer Review file · Nature Communications]

Support-Tuned Iridium Reconstruction with Crystalline Phase Dominating Acidic Oxygen Evolution

Corresponding Author: Professor Xiaoxin Zou

Version 0:

Reviewer comments:

Reviewer #1

(Remarks to the Author)

In this work, the authors propose that the core@shell TiOx@Ti substrate induces the complete bulk phase transition from metallic Ir to thermodynamically stable crystalline rutile IrO2. Combined with a series of advanced tests and theoretical calculations, the authors prove that this support-guided crystallization transfers the dominant OER mechanism from LOM to AEM. The Ir/TiOx@Ti exhibits excellent activity and durability in the three-electrode system and proton exchange membrane water electrolyzers. The efforts made are highly commendable, and this study may be useful to researchers in the field of Ir-based catalysts for acidic OER. The manuscript can be publishable after the revision as described below:

1. The authors claim that the TiOx@Ti support induces the complete bulk phase transition of metallic Ir to thermodynamically stable crystalline rutile IrO2, but the authors do not give sufficient explanation and evidence as to what role the TiOx@Ti support plays in this process. In addition, the TiOx@Ti support in this work is a core-shell structure, so what is the significance of this structure for the support-guided phase transition?
2. As mentioned by the authors, IrO2 has a more thermodynamically stable structure than metallic Ir, so why not directly load IrO2 on the support to catalyze acidic OER, and choose the transition from metallic Ir to IrO2?
3. In situ Ir L3-edge XANES spectra, the potential of Ir/TiOx@Ti is from OCP to 1.65 V, and Ir NPs is from OCP to 1.45 V. Is it fair to judge the oxidation of Ir in the two samples from this?
4. It is recommended to supplement microscopic morphology or phase structure analysis of the active species and support in Ir/TiOx@Ti after durability testing.
5. Supplementary Figure 6 shows that the XRD patterns of Ir/TiOx@Ti with different Ir content are essentially the same as those of Ti nanospheres without any of the diffraction peaks of Ir (PDF#06-0598), which is not consistent with the Ir NPs shown in TEM images (Supplementary Figure 7).
6. In Supplementary Table 2, the Ir-O coordination number in Ir NPs increases from 2.8 to 4.1 when the potential changes from OCP to 1.45 V, but it is described in the manuscript as 3.1 to 4.1 (line 217).
7. In the MEA preparation, the authors indicate that the anode Ir loading is 0.3 mg/cm2, but in Supplementary Table 4, the Ir loading of Ir/TiOx@Ti is described as 0.28 mg/cm2.
8. In Equation (1) in line 427, Evs. SCE is misspelled as Evs. SHE. The authors need to thoroughly check the manuscript for possible errors and correct them.

Reviewer #2

(Remarks to the Author)

The authors are fighting an "invisible giant", since the point does not even exist. In industry, Ir black operates in PEMWE, with a decay rate as low as 50 $\mu\text{V/h}$ (e.g. commercial product at CONTANGO). The commercial IrOx even has the stability at <10 $\mu\text{V/h}$. Herein, it would be interesting to see the authors' comparison trying to make the case. Not even mentioning that in real case of PEMWE, the supporting materials are TiOx/Ti.

The story is really nice. But, do I buy it? The answer would be no. The data quality is not high enough to justify. Since Ir forms very thin IrOx layer during OER, the valence comparison in Fig.4g is not really fair. 4b vs.4c, may just come from the different content of Ir. TEM analysis in Fig5 is not solid, since I may get the same data set by just tilting the sample a tiny bit. In short, I don't see a ground making the story believable.

Bottom line, the Ir/TiO_x herein is impressive in low Ir-content and activity. After the unfair comparison is removed, I am sure the manuscript can be published somewhere.

Reviewer #3

(Remarks to the Author)

In this manuscript, Zhang et al. have demonstrated how the TiO_x@Ti support has tuned the reconstruction of Ir nanoparticles to make them active and stable for acidic OER. The manuscript is well-written in good structure, with detailed characterization and analysis. However, before publication, there are some issues to be addressed.

First is about the reason for phase transition into IrO₂. Of course, the Ti support plays an important role in the process. However, it is also important to clear point out the influence of nanoparticle size. For example, it has been revealed the particle size of CoO_x influences the Co oxidation and reconstruction (Nat Energy 7, 765–773 (2022)). In this manuscript, it is better to provide the statistics of Ir size without the support, to compare the results in Figure 2e. A comparative discussion on this could avoid the misleading to the readers.

Second, the characterizations and analysis about the AEM and LOM pathways in Figure 6 are in detail and impressive. However, it is a bit confusing in the discussion and the conclusion. If just looking at the data presented in Figure 6c and Supplementary Figure 18, it is difficult to imagine that the LOM is dominant in any of the catalysts here, since the 34O₂/32O₂ ratios are as low as 0.4% to 0.6%. Therefore, it is more likely the AEM is always the dominated pathway. There could be related literature to support this point. On the other hand, the authors are right about that in Ir/TiO_x@Ti, the LOM (even it is not dominated) is suppressed compared to the unsupported Ir. Instead of saying the pathway is shifted from LOM from AEM, it more appropriate to say that even the minor LOM pathway is suppressed in Ir/TiO_x@Ti.

Third, in Figure 4f-g, it is unclear how the change in the white line of Ir L edge is converted into the Ir oxidation state. The reference spectra of metallic Ir and IrO₂ are also not provided. Please provide the analysis and discussion in detail.

The change of coordination number at the Ir-O bond can be plotted as a function of applied potential, to compare the difference between two samples. In fact, this information could be even more important than the present Figure 4h-i.

Other comments:

The error bar in Figure 3b needs to be defined.

The electrochemical conditions for the 10-h ICP measurement are not clear, CP, CA or others?

On page 4 line 78, the 'core@shell TiO_x@Ti' is wrong, it should be 'shell@core'.

On page 9 line 187, "metallic surface" could be confusing. The authors tried to say that the metallic Ir is changed into Ir oxides. However, metallic surface could sometimes be interpreted as the surface has the metallic conductivity. Therefore, it should be defined better to avoid misleading to the authors.

Reviewer #4

(Remarks to the Author)

Using an extensive combination of experimental imaging and spectroscopic approaches supported by DFT calculations, the authors show that when deposited on a core-shell Ti/TiO₂ nanoparticle, Ir undergoes a complete structural transformation to IrO₂ rather than the usual surface oxidation. This has markedly improves both the stability and the OER activity of the catalytically active Ir/IrO₂ particles. Finally long-term stable and active operation in an electrolyzer device is demonstrated.

While I believe the work is already of high quality, I believe some points need to be revisited to reach the level required for publication in Nature Communications.

1) First and foremost, the authors repeatedly invoke metal-support interactions (or similar terms) to explain the observed effects. It, however, remains unclear until the end what exactly the role of the TiO₂ support is and if the Ti core plays any role. The model is based on a transformation from Ir(110) to IrO₂(211), which appear to have compatible lattices. Yet, what is the support doing? Is it providing strain that eases the O incorporation into Ir? Does it supply oxygen atoms? If so, what are the oxygen chemical potentials in TiO₂ and IrO₂? It would significantly strengthen the message of the manuscript if the authors could provide a plausible explanation and back it up with experimental or computational data.

2) Why is IrO₂ so much more stable against dissolution than IrO_x? Is the support playing a role here other than easing the transformation? How does this agree with the claim of "... mitigating dissolution through strong metal-support interactions." on page 8 of the manuscript? Also, the LOM/AEM crossover already occurs before reaching IrO. How is this reconciled with the higher stability for the completely transformed structure?

3) Does the transformation to IrO₂ in any way depend on the size of the Ir particles? Can particles of any size be completely transformed?

4) Could the authors detail how the IrO_{0.25}, IrO_{0.5} and IrO models used for DFT analysis were constructed? How can the structural relevance of the contained active sites be ascertained?

5) It is not entirely clear from the data how to authors arrive at the (211) surface orientation from the Fourier transforms.

6) For the DFT calculations, what semi-core states were included for the metal atoms? Is the k-sampling sufficient for metals? From where are the ZPE and entropy corrections taken?

7) In figure S4, how is the energy difference defined? Does smaller imply an easier migration into the lattice?

Version 1:

Reviewer comments:

Reviewer #3

(Remarks to the Author)

My comments have been properly addressed with the revision, and therefore I recommend publishing the manuscript in Nature Communications.

Reviewer #4

(Remarks to the Author)

The authors have addressed my previous queries and amended the manuscript, improving clarity and reproducibility. I recommend acceptance of the manuscript in its present form.

Reviewer #5

(Remarks to the Author)

Electrochemical water splitting has become an attractive alternative clean and efficient hydrogen production method. To this end, the fabrication of low-cost, stable and high-efficiency electrocatalysts for water electrolysis is crucial. In this paper, the authors showed the fabrication of Ir/TiO_x@Ti catalyst for OER with high activity and high durability in acidic media. However, the novel opinions are difficult to be found. The following questions should be addressed before publication in other journal.

1. For TiO_x@Ti, what is the value of x? What impact does x value on the catalytic performance of Ir?
2. In TiO_x@Ti, is the thickness of amorphous TiO_x shell controllable? What are the effects of different TiO_x shell thicknesses on the catalytic performance of Ir? What is the optimal thickness of TiO_x shell?
3. The authors think the amorphous TiO_x shell consists of Ti₃O₅/Ti₄O₇-like clusters. Is the molar ratio of Ti₃O₅/Ti₄O₇ in the amorphous TiO_x shell controllable? What are the effects of different molar ratios of Ti₃O₅/Ti₄O₇ on the catalytic performance of Ir?
4. In Figure 3a and 3b, what is the percentage of Ir content? Is it mass percentage content or molar percentage content? Please indicate clearly.
5. For Ir/TiO_x@Ti catalysts, is Ir completely converted to crystalline IrO₂ during the catalytic process? Why haven't you discussed achieving the optimal ratio of IrO₂/Ir?

Version 2:

Reviewer comments:

Reviewer #5

(Remarks to the Author)

The author provided a good response to my concerns, and I suggest accepting this paper.

吉林大学 化学学院

Department of Chemistry

Jilin University

Changchun 130012, China

Dr. Prof. Xiaoxin Zou

State Key Lab. Inorg. Synth. & Prep. Chem.

Tel: +86-431-85168221

xxzou@jlu.edu.cn

<http://zouxxgroup.com/>

Reviewer's Comments and Our Responses:

Reviewer #1:

In this work, the authors propose that the core@shell $\text{TiO}_x\text{@Ti}$ substrate induces the complete bulk phase transition from metallic Ir to thermodynamically stable crystalline rutile IrO_2 . Combined with a series of advanced tests and theoretical calculations, the authors prove that this support-guided crystallization transfers the dominant OER mechanism from LOM to AEM. The Ir/ $\text{TiO}_x\text{@Ti}$ exhibits excellent activity and durability in the three-electrode system and proton exchange membrane water electrolyzers. The efforts made are highly commendable, and this study may be useful to researchers in the field of Ir-based catalysts for acidic OER. The manuscript can be publishable after the revision as described below:

Response: We sincerely appreciate the Reviewer's insightful evaluation and positive feedback on our work. The reviewer has provided an excellent overview of our research focus and offered valuable suggestions for improvement. We have carefully addressed all the revision suggestions raised by the reviewer and have incorporated corresponding modifications into the revised manuscript to enhance clarity and rigor.

1. The authors claim that the $\text{TiO}_x\text{@Ti}$ support induces the complete bulk phase transition of metallic Ir to thermodynamically stable crystalline rutile IrO_2 , but the authors do not give sufficient explanation and evidence as to what role the $\text{TiO}_x\text{@Ti}$ support plays in this process. In addition, the $\text{TiO}_x\text{@Ti}$ support in this work is a core-shell structure, so what is the significance of this structure for the support-guided phase transition?

Response 1: We sincerely appreciate the reviewer's valuable comments. Below, we provide detailed clarifications on each question raised.

Our response to the reviewer's question (1): the authors do not give sufficient explanation and evidence as to what role the $\text{TiO}_x\text{@Ti}$ support plays in this process.

To elucidate the role of the $\text{TiO}_x\text{@Ti}$ support in Ir oxidation, we employed *in situ* ^{18}O isotope labeling experiments to identify the oxygen source for Ir oxidation. The $\text{Ti}^{16}\text{O}_x\text{@Ti}$ sample was first employed to catalytic reaction in ^{18}O -labeled H_2O for 10 h, yielding $\text{IrO}_2/\text{Ti}^{16}\text{O}_x\text{@Ti}$. In this process, the oxygen incorporated into the Ir lattice could originate from two potential pathways: (i) oxygen derived from water dissociation or (ii) oxygen migration from the oxide support. To determine the oxygen source in the formed $\text{IrO}_2/\text{Ti}^{16}\text{O}_x\text{@Ti}$, we conducted thermogravimetric analysis coupled with mass spectrometry (TG-MS). During this analysis, IrO_2 decomposes into metallic Ir and O_2 when heated to 1000 °C under an Ar atmosphere. The TG-MS results unambiguously showed that the evolved O_2 consisted exclusively of $^{18}\text{O}_2$ (Supplementary Figure 22), confirming that the oxygen incorporated into the Ir lattice originated solely from water

molecules. This finding, combined with CV and *in situ* XAS results (**Figure 4**), demonstrates that the TiO_x@Ti support facilitates both oxygen covering on Ir surface and oxygen bulk diffusion in Ir lattice.

Understanding of metal oxidation mechanisms at the atomic scale, particularly regarding oxygen diffusion pathways and metal lattice evolution, remains fundamentally challenging. The electronic interactions between TiO_x@Ti and Ir nanoparticles may play a non-negligible role in promoting the oxidation of iridium. To probe these interfacial electronic effects, we conducted work function measurements, which revealed values of 4.8 eV for TiO_x@Ti and 5.4 eV for Ir nanoparticles (Supplementary Figure 23). This measurable difference in work functions establishes the thermodynamic driving force for electron transfer from the TiO_x@Ti support to the supported Ir nanoparticles. Compared to unsupported Ir NPs, this electron transfer appears to weaken the originally strong Ir-O bonds by populating their antibonding orbitals (*Nat. Commun.*, **2024**, *15*, 1780; *ACS Appl. Mater. Interfaces*, **2018**, *10*, 38117-38124), thereby could facilitate oxygen incorporation into the Ir lattice during OER. These observations provide a possible electronic-structure-based explanation for the enhanced oxidation kinetics observed in our supported catalyst.

The corresponding discussion has been incorporated into the **Figure 5** section of the revised manuscript.

Our response to the reviewer's question (2): what is the significance of this core-shell structure for the support-guided phase transition?

Having previously discussed the unique role of the TiO_x@Ti support in inducing the exceptional phase transformation of Ir, we now address the reviewer's further inquiry into the role of the metallic Ti core. Considering that the TiO_x layer on the Ti contains Ti₄O₇-like clusters, we synthesized Ti₄O₇ (without a metallic Ti core) as a control material. For direct comparison, Ir nanoparticles supported on Ti₄O₇ were synthesized using the same method, achieving uniform and dense deposition on the Ti₄O₇ surface. Post-OER HAADF-STEM analysis revealed that Ir on Ti₄O₇, similar to unsupported Ir NPs, undergoes significant surface amorphization (Supplementary Figure 21). This stark contrast underscores the critical role of the amorphous TiO_x layer derived from metallic Ti in driving the phase transformation, a feature absent in crystalline Ti₄O₇-supported systems. We tentatively propose that the unique core-shell architecture of TiO_x@Ti might influence the interfacial electronic structure in ways that could facilitate Ir oxidation. The relatively low work function of metallic Ti (4.3 eV, *Appl. Phys. Lett.*, **2009**, *95*), compared to its oxide counterparts, may establish a graded electron transfer pathway from Ti to TiO_x to Ir. Such electronic configuration could potentially weaken the oxygen-binding ability, thereby possibly enhancing oxygen diffusion within the Ir lattice. However, we acknowledge that further investigations would be required to fully elucidate these complex interfacial phenomena.

The corresponding discussion has been incorporated into the **Figure 5** section of the revised manuscript.

2. As mentioned by the authors, IrO₂ has a more thermodynamically stable structure than metallic

Ir, so why not directly load IrO₂ on the support to catalyze acidic OER, and choose the transition from metallic Ir to IrO₂?

Response 2: We appreciate reviewer's insightful question. Our considerations primarily address two aspects: (1) It is widely recognized that metallic Ir-derived IrO_x generally exhibit superior activity compared to crystalline IrO₂ but suffer from inferior stability (*Energy Environ. Sci.*, **2025**, *18*, 1214-1231; *ACS Catal.*, **2019**, *9*, 4688-4698). In this work, through phase transformation engineering, we enhanced the structural stability of metallic Ir-derived IrO₂ while achieving synergistic optimization of activity and stability. Notably, the derived IrO₂ with a unique (211) facet demonstrates higher intrinsic activity than conventional IrO₂(110) surfaces. (2) The emphasis of our work lies not in highlighting exceptional activity or stability but in revealing a novel mechanism: the TiO_x@Ti support induces complete bulk oxidation of metallic Ir to IrO₂. This discovery redefines the reconstruction paradigm in Ir oxidation by establishing catalyst supports as dynamic architects of active phases.

3. In situ Ir L3-edge XANES spectra, the potential of Ir/TiO_x@Ti is from OCP to 1.65 V, and Ir NPs is from OCP to 1.45 V. Is it fair to judge the oxidation of Ir in the two samples from this?

Response 3: We have revised **Figure 4g** in the updated manuscript to directly compare the valence state evolution of Ir/TiO_x@Ti and unsupported Ir NPs under identical voltage conditions, ensuring a more objective and equitable analysis. The results demonstrate that Ir on Ir/TiO_x@Ti undergoes significantly faster oxidation compared to unsupported Ir NPs.

4 It is recommended to supplement microscopic morphology or phase structure analysis of the active species and support in Ir/TiO_x@Ti after durability testing.

Response 4: We appreciate the reviewer's valuable suggestion. To reinforce our conclusions and strengthen the data rigor, we have supplemented additional high-resolution images of Ir/TiO_x@Ti at varying catalytic durations (Supplementary Figure 17) and the XRD pattern of Ir/TiO_x@Ti after 10 hours of catalysis (Supplementary Figure 16). The XRD pattern after catalysis reveals that the sample retains its original structure, displaying only the characteristic diffraction peaks of metallic Ti, with no detectable IrO₂ signals—consistent with the small-size effect of Ir. Crucially, HAADF-STEM precisely captured the structural transformation process from metallic Ir to IrO₂, providing compelling evidence for the oxidation pathway.

5 Supplementary Figure 6 shows that the XRD patterns of Ir/TiO_x@Ti with different Ir content are essentially the same as those of Ti nanospheres without any of the diffraction peaks of Ir (PDF#06-0598), which is not consistent with the Ir NPs shown in TEM images (Supplementary Figure 7).

Response 5: The lack of distinct diffraction peaks for the Ir nanoparticles is primarily attributed to their extremely small size (~2 nm), which leads to significant peak broadening and reduced

intensity due to the Scherrer effect. In contrast, the Ti nanospheres, with a much larger size (~60 nm) and a well-crystalline core, exhibit sharp and intense diffraction peaks. Furthermore, the strongest diffraction peak of the Ir nanoparticles (~41°) closely overlaps with the dominant peak of the Ti nanospheres (~40°). This overlap, combined with the overwhelming signal intensity from the Ti phase, results in the masking of Ir's diffraction features in Ir/TiO_x@Ti's XRD. Critically, a series of characterization techniques in the manuscript have confirmed the successful deposition of Ir NPs on the TiO_x@Ti support.

6 In Supplementary Table 2, the Ir-O coordination number in Ir NPs increases from 2.8 to 4.1 when the potential changes from OCP to 1.45 V, but it is described in the manuscript as 3.1 to 4.1 (line 217).

Response 6: We apologize for the typographical error. The coordination number of Ir NPs at OCP was mistakenly described as the value at 1.25 V. This has been corrected to the accurate value of 2.8 in the revised version.

7 In the MEA preparation, the authors indicate that the anode Ir loading is 0.3 mg/cm², but in Supplementary Table 4, the Ir loading of Ir/TiO_x@Ti is described as 0.28 mg/cm².

Response 7: The MEA testing results have been validated through multiple tests, with XRF measurements indicating that the loading fluctuates within the range of 0.28–0.31 mg/cm². Therefore, we adopted a loading value of 0.3, which is close to the average, as the actual loading. Accordingly, we corrected this value to 0.3 in the revised Supplementary Table 4 and implemented corresponding adjustments in **Figure 7c** to ensure data consistency.

8 In Equation (1) in line 427, E vs. SCE is misspelled as E vs. SHE. The authors need to thoroughly check the manuscript for possible errors and correct them.

Response 8: We sincerely appreciate the reviewer's meticulous and conscientious evaluation. We apologize for the spelling errors, which have been corrected in the revised manuscript. Additionally, we have conducted a thorough review of the entire manuscript to ensure accuracy.

Reviewer #2:

The authors are fighting an "invisible giant", since the point does not even exist. In industry, Ir black operates in PEMWE, with a decay rate as low as 50 μV/h (e.g. commercial product at CONTANGO). The commercial IrO_x even has the stability at <10 μV/h. Herein, it would be interesting to see the authors' comparison trying to make the case. Not even mentioning that in real case of PEMWE, the supporting materials are TiO_x/Ti.

The story is really nice. But, do I buy it? The answer would be no. The data quality is not high enough to justify. Since Ir forms very thin IrO_x layer during OER, the valence comparison in Fig.4g is not really fair. 4b vs. 4c, may just come from the different content of Ir. TEM analysis in

Fig.5 is not solid, since I may get the same data set by just tilting the sample a tiny bit. In short, I don't see a ground making the story believable.

Bottom line, the Ir/TiO_x herein is impressive in low Ir-content and activity. After the unfair comparison is removed, I am sure the manuscript can be published somewhere.

Response: We sincerely appreciate the Reviewer's partial recognition of our work, particularly the statement that "the Ir/TiO_x herein is impressive in low Ir-content and activity." This encouraging feedback motivates us to refine our work further. We deeply value the reviewer's critical comments, which have significantly improved the quality of our manuscript. To address the concerns raised, we have meticulously revised the manuscript to enhance its accuracy and scientific rigor. We sincerely hope that the reviewer will find our revised manuscript significantly improved and kindly request reconsideration of our submission. We greatly appreciate your time and effort in reviewing our work.

Our response to the reviewer's comment (1): The authors are fighting an "invisible giant"... Not even mentioning that in real case of PEMWE, the supporting materials are TiO_x/Ti.

We fully acknowledge the excellent stability of commercial Ir black (e.g., CONTANGO's Ir black) and IrO_x (typically rutile IrO₂). However, our work focuses on low-Ir-loading CCM (0.3 mg/cm²), where conventional catalysts face severe challenges. Commercial Ir black typically achieves stability under high Ir loadings (e.g., 1 mg/cm²; data source: <https://www.contango.cn/h-col-127.html>). However, its performance at low Ir loadings remains uncertain. To directly validate this, we tested CONTANGO Ir black (with ~6 nm Ir particles, see Supplementary Figure 32) under identical MEA conditions at a low Ir loading of 0.3 mg/cm² (Supplementary Figure 33). Its MEA activity reached 3.2 A/cm² at 2 V, significantly lower than Ir/TiO_x@Ti (4.0 A/cm² at 2 V). Critically, its decay rate at 2 A/cm² was 315 μV/h, far exceeding the 52 μV/h for Ir/TiO_x@Ti. This confirms the superiority of Ir/TiO_x@Ti over commercial Ir black at low Ir loading.

The reviewer's observation regarding conventional TiO_x supports in PEMWEs is well-noted. While TiO_x substrates are widely used in PEMWEs due to their structural stability and corrosion resistance, these typically refer to well-crystalline TiO₂ supporting IrO₂ with high Ir loadings (>40 wt%) to compensate for TiO₂'s poor conductivity. To validate the advantage of Ir/TiO_x@Ti over commercial TiO₂-supported catalysts, we tested Heraeus's commercial IrO₂/TiO₂ (S60, with ~2 nm IrO₂ particles supported on ~30 nm crystalline TiO₂; see Supplementary Figure 32) at 0.3 mg/cm² Ir loading (Supplementary Figure 33). Its MEA activity reached 2.9 A/cm² at 2 V, inferior to Ir/TiO_x@Ti (4.0 A/cm²), and its decay rate at 2 A/cm² was 185 μV/h, still significantly higher than Ir/TiO_x@Ti's 52 μV/h. This further highlights the superiority of Ir/TiO_x@Ti at low Ir loading.

While the comparative analyses presented above conclusively demonstrate the superior performance of Ir/TiO_x@Ti, we would like to emphasize that the core innovation of this work lies not solely in pursuing high-performance Ir catalysts but in uncovering an unconventional phenomenon: the TiO_x@Ti support enables electrochemical conversion of metallic Ir into crystalline rutile IrO₂—a stark contrast to the conventional understanding that Ir typically form amorphous IrO_x. This support-induced phase transition strategy enables synergistic improvements in activity and stability of electrochemically oxidized Ir species, underscoring the role of supports

as dynamic architects of active phases. We believe this insight will invigorate research on Ir oxidation mechanisms and inspire the development of high-performance Ir-derived catalysts.

Our response to the reviewer's comment (2): Since Ir forms very thin IrO_x layer during OER, the valence comparison in Fig. 4g is not really fair.

XAS-based valence state fitting is widely employed to characterize the average valence state of Ir-based catalysts, *i.e.*, valence states in Fig. 4g represent an average of bulk valence (not just surface oxidation), making it reasonable to compare the overall oxidation between Ir/TiO_x@Ti and Ir NPs. As shown in Fig. 4g, Ir/TiO_x@Ti reached an average valence state of +3.2 (approaching IrO₂'s +4) at 1.45 V, far exceeding that of Ir NPs (+2.0). To enhance clarity, we have revised Fig. 4g to explicitly highlight the stark contrast in oxidation extent between the two materials under identical potentials.

Our response to the reviewer's comment (3): 4b vs. 4c may stem from different Ir content.

We wish to clarify that all electrochemical tests in Fig. 4b–e were conducted with a fixed Ir loading of 0.18 mg/cm² on the working electrode to eliminate content-related biases. This clarification has been added to the Fig. 4's caption in the revised manuscript.

Our response to the reviewer's comment (4): Fig. 5 is not solid, since I may get the same data set by just tilting the sample a tiny bit.

HAADF-STEM has now become a well-established technique for high-precision characterization of atomic-resolution local structures. Although crystal tilting may introduce minor lattice shifts, the impact of crystal tilting is on the order of 10 pm (0.1 Å), and more severe tilting would result in directional blurring of images rather than systematic lattice shifts—such images can be readily identified and excluded during analysis (*J. Electron Microsc.*, **2012**, *61*, 207-215). In our STEM analysis, the lattice spacing changes observed during the transformation process (from initial to final states) reached up to 0.3 Å, which clearly cannot be attributed to tilting artifacts but instead confirms the occurrence of a genuine phase transition.

To further strengthen our data, eliminate measurement uncertainties, and enhance rigor, we have provided three additional independent TEM datasets for each catalytic time-point sample in Supplementary Figure 17. The results demonstrate minimal distance errors (<0.04 Å) across parallel STEM images.

Reviewer #3:

In this manuscript, Zhang *et al.* have demonstrated how the TiO_x@Ti support has tuned the reconstruction of Ir nanoparticles to make them active and stable for acidic OER. The manuscript is well-written in good structure, with detailed characterization and analysis. However, before publication, there are some issues to be addressed.

Response: We sincerely appreciate the Reviewer's thorough evaluation and constructive feedback

on our work. We appreciate the reviewer's acknowledgment of the manuscript's structured presentation and detailed characterization. All concerns raised by the reviewer have been carefully addressed in the revised manuscript. We have incorporated corresponding modifications to enhance the clarity and rigor. These adjustments have been systematically integrated into the text, figures, and supplementary materials to ensure a cohesive and robust presentation of our findings.

1. First is about the reason for phase transition into IrO₂. Of course, the Ti support plays an important role in the process. However, it is also important to clear point out the influence of nanoparticle size. For example, it has been revealed the particle size of CoO_x influences the Co oxidation and reconstruction (Nat Energy 7, 765–773 (2022)). In this manuscript, it is better to provide the statistics of Ir size without the support, to compare the results in Figure 2e. A comparative discussion on this could avoid the misleading to the readers.

Response 1: We appreciate the reviewer's valuable comment. We agree that particle size may influence this transition. However, based on our statistical analysis, the average size of unsupported Ir NPs is 2.1 nm, which is nearly identical to the 2.0 nm Ir particle size on Ir/TiO_x@Ti. Therefore, in our work, we conclude that particle size is not responsible for the differing transformation pathways between unsupported Ir NPs and Ir/TiO_x@Ti; instead, the support plays a critical role. We have added the particle size distribution of unsupported Ir NPs in Supplementary Figure 9 and specified the 2.1 nm Ir particle size in the revised manuscript.

2. Second, the characterizations and analysis about the AEM and LOM pathways in Figure 6 are in detail and impressive. However, it is a bit confusing in the discussion and the conclusion. If just looking at the data presented in Figure 6c and Supplementary Figure 18, it is difficult to imagine that the LOM is dominant in any of the catalysts here, since the ³⁴O₂/³²O₂ ratios are as low as 0.4% to 0.6%. Therefore, it is more likely the AEM is always the dominated pathway. There could be related literature to support this point. On the other hand, the authors are right about that in Ir/TiO_x@Ti, the LOM (even it is not dominated) is suppressed compared to the unsupported Ir. Instead of saying the pathway is shifted from LOM from AEM, it more propriate to say that even the minor LOM pathway is suppressed in Ir/TiO_x@Ti.

Response 2: We appreciate the reviewer's constructive feedback. We acknowledge that the original statement claiming "LOM dominance" might lead to misinterpretation. Accordingly, we have revised all expressions in the manuscript such as "shifts the dominant OER mechanism from LOM to AEM" to "shifts the reaction mechanism from LOM-participated to the complete AEM" to enhance scientific rigor. Nevertheless, we wish to clarify that although the participation level of LOM in OER is typically low, even minimal involvement can inflict profound detrimental effects on structural integrity (Nat. Sci. Rev., 2024, 11, nwae362; ACS Catal., 2018, 8, 9765-9774). Therefore, the strategic suppression of LOM pathways remains critically important for enhancing both structural and catalytic stability.

3. Third, in Figure 4f-g, it is unclear how the change in the white line of Ir L edge is converted into the Ir oxidation state. The reference spectra of metallic Ir and IrO₂ are also not provided. Please provide the analysis and discussion in detail.

Response 3: The valence state fitting for XANES was performed by establishing a linear relationship between the white-line peak intensity of the Ir L₃-edge and reference standards (metallic Ir and IrO₂), following the method described in *J. Am. Chem. Soc.*, **2021**, *143*, 12524-12534. To address this, we have added clarifying remarks in the caption of **Figure 4g**, included the reference spectra for metallic Ir and IrO₂ in Supplementary Figure 14, and supplemented the Methods section with a description of the fitting method to resolve ambiguities.

4. The change of coordination number at the Ir-O bond can be plotted as a function of applied potential, to compare the difference between two samples. In fact, this information could be even more important than the present Figure 4h-i.

Response 4: We appreciate the reviewer's insightful suggestion. We have replotted **Figure 4g** to explicitly compare the evolution of Ir-O coordination numbers as a function of applied potential between the two samples. The results confirm that Ir/TiO_x@Ti undergoes deeper bulk oxidation compared to unsupported Ir.

5. Other comments:

The error bar in Figure 3b needs to be defined.

Response 5: In the revised manuscript, the error bars in **Figure 3b** are now explicitly defined as representing the standard deviations of three independent measurements.

6. The electrochemical conditions for the 10-h ICP measurement are not clear, CP, CA or others?

Response 6: The 10-hour ICP measurement was conducted under a constant potential of 1.55 V vs. RHE (*i.e.*, chronoamperometry, CA). This clarification has been added to the caption of **Figure 3c** in the revised manuscript.

7. On page 4 line 78, the 'core@shell TiO_x@Ti' is wrong, it should be 'shell@core'.

Response 7: We appreciate the reviewer's meticulous observation. All instances of "core@shell TiO_x@Ti" in the manuscript have been corrected to "shell@core" TiO_x@Ti.

8. On page 9 line 187, "metallic surface" could be confusing. The authors tried to say that the metallic Ir is changed into Ir oxides. However, metallic surface could sometimes be interpreted as the surface has the metallic conductivity. Therefore, it should be defined better to avoid misleading to the authors.

Response 8: We fully agree with the reviewer's point. In the revised manuscript, the term "metallic surface" has been revised to "metallic surface characteristics" to avoid potential misinterpretation.

Reviewer #4:

Using an extensive combination of experimental imaging and spectroscopic approaches supported by DFT calculations, the authors show that when deposited on a core-shell Ti/TiO₂ nanoparticle, Ir undergoes a complete structural transformation to IrO₂ rather than the usual surface oxidation. This has markedly improves both the stability and the OER activity of the catalytically active Ir/IrO₂ particles. Finally long-term stable and active operation in an electrolyzer device is demonstrated.

While I believe the work is already of high quality, I believe some points need to be revisited to reach the level required for publication in Nature Communications.

Response: We sincerely appreciate the Reviewer's thorough evaluation of our work and recognition of the integrated experimental-computational approach used to elucidate the complete structural transformation of Ir on the TiO_x@Ti support. We are grateful for the reviewer's positive assessment of the enhanced stability, OER activity, and practical electrolyzer performance demonstrated in this study. All concerns raised by the reviewer have been carefully addressed in the revised manuscript. Corresponding adjustments have been implemented in the text and supplementary materials to ensure methodological rigor and clarity. We thank the reviewer for his/her constructive suggestions, which have significantly improved the scientific depth and coherence of the manuscript.

1) First and foremost, the authors repeatedly invoke metal-support interactions (or similar terms) to explain the observed effects. It, however, remains unclear until the end what exactly the role of the TiO₂ support is and if the Ti core plays any role. The model is based on a transformation from Ir(110) to IrO₂(211), which appear to have compatible lattices. Yet, what is the support doing? Is it providing strain that eases the O incorporation into Ir? Does it supply oxygen atoms? If so, what are the oxygen chemical potentials in TiO₂ and IrO₂? It would significantly strengthen the message of the manuscript if the authors could provide a plausible explanation and back it up with experimental or computational data.

Response 1: We appreciate the reviewer's constructive comments. We acknowledge that the discussion on the role of the support and Ti core was insufficient in the original manuscript. Accordingly, we have supplemented additional experiments and expanded the discussion in **Figure 5** of the revised manuscript.

Our response to the role of the support in Ir oxidation:

To elucidate the role of the TiO_x@Ti support in Ir oxidation, we employed *in situ* ¹⁸O isotope labeling experiments to identify the oxygen source for Ir oxidation. The Ti¹⁶O_x@Ti sample was

first employed to catalytic reaction in ^{18}O -labeled H_2O for 10 h, yielding $\text{IrO}_2/\text{Ti}^{16}\text{O}_x@\text{Ti}$. In this process, the oxygen incorporated into the Ir lattice could originate from two potential pathways: (i) oxygen derived from water dissociation or (ii) oxygen migration from the oxide support. To determine the oxygen source in the formed $\text{IrO}_2/\text{Ti}^{16}\text{O}_x@\text{Ti}$, we conducted thermogravimetric analysis coupled with mass spectrometry (TG-MS). During this analysis, IrO_2 decomposes into metallic Ir and O_2 when heated to 1000 °C under an Ar atmosphere. The TG-MS results unambiguously showed that the evolved O_2 consisted exclusively of $^{18}\text{O}_2$ (Supplementary Figure 22), confirming that the oxygen incorporated into the Ir lattice originated solely from water molecules. This finding, combined with CV and *in situ* XAS results (**Figure 4**), demonstrates that the $\text{TiO}_x@\text{Ti}$ support facilitates both surface O covering and bulk O diffusion in Ir lattice.

Understanding of metal oxidation mechanisms at the atomic scale, particularly regarding oxygen diffusion pathways and metal lattice evolution, remains fundamentally challenging. The electronic interactions between $\text{TiO}_x@\text{Ti}$ and the Ir may play a non-negligible role in promoting the oxidation of iridium. To probe these interfacial electronic effects, we conducted work function measurements, which revealed values of 4.8 eV for $\text{TiO}_x@\text{Ti}$ and 5.4 eV for Ir nanoparticles (Supplementary Figure 23). This measurable difference in work functions establishes the thermodynamic driving force for electron transfer from the $\text{TiO}_x@\text{Ti}$ support to the supported Ir nanoparticles. Compared to unsupported Ir NPs, this electron transfer appears to weaken the originally strong Ir-O bonds by populating their antibonding orbitals (*Nat. Commun.*, **2024**, *15*, 1780; *ACS Appl. Mater. Interfaces*, **2018**, *10*, 38117-38124), thereby could promote oxygen incorporation into the Ir lattice during OER. These observations provide a possible electronic-structure-based explanation for the enhanced oxidation kinetics observed in our supported catalyst.

As outlined above, our findings demonstrate that the $\text{TiO}_x@\text{Ti}$ support does not induce Ir phase transformation by supplying oxygen. Instead, it could facilitate oxygen incorporation into the Ir lattice *via* electron transfer from the support to the Ir nanoparticles. This mechanism is further supported by the observation that the pristine Ir lattice in $\text{Ir}/\text{TiO}_x@\text{Ti}$ (**Figures 2e** and **5d**) perfectly matches the lattice parameters of standard metallic Ir, confirming no initial strain is imposed by the support. Thus, the phase transformation may be driven by electronic interactions at the Ir-support interface rather than mechanical strain or direct oxygen supply from the support.

Our response to the role of the Ti core:

Having previously discussed the unique role of the $\text{TiO}_x@\text{Ti}$ support in inducing the exceptional phase transformation of Ir, we now address the reviewer's further inquiry into the role of the metallic Ti core. Considering that the TiO_x layer on the Ti contains Ti_4O_7 -like clusters, we synthesized Ti_4O_7 (without a metallic Ti core) as a control material. For direct comparison, Ir nanoparticles supported on Ti_4O_7 were synthesized using the same method, achieving uniform and dense deposition on the Ti_4O_7 surface. Post-OER HAADF-STEM analysis revealed that Ir on Ti_4O_7 , similar to unsupported Ir NPs, undergoes significant surface amorphization (Supplementary Figure 21). This stark contrast underscores the critical role of the amorphous TiO_x layer derived from metallic Ti in driving the phase transformation, a feature absent in crystalline Ti_4O_7 -supported systems. We tentatively propose that the unique core-shell architecture of $\text{TiO}_x@\text{Ti}$ might

influence the interfacial electronic structure in ways that could facilitate Ir oxidation. The relatively low work function of metallic Ti (4.3 eV, *Appl. Phys. Lett.*, **2009**, 95), compared to its oxide counterparts, may establish a graded electron transfer pathway from Ti to TiO_x to Ir. Such electronic configuration could potentially weaken the iridium-oxygen binding, thereby possibly enhancing oxygen diffusion within the Ir lattice. However, we acknowledge that further investigations would be required to fully elucidate these complex interfacial phenomena.

2) Why is IrO₂ so much more stable against dissolution than IrO_x? Is the support playing a role here other than easing the transformation? How does this agree with the claim of "... mitigating dissolution through strong metal-support interactions." on page 8 of the manuscript? Also, the LOM/AEM crossover already occurs before reaching IrO. How is this reconciled with the higher stability for the completely transformed structure?

Response 2: Our response to the reviewer's question (1): Why is IrO₂ so much more stable against dissolution than IrO_x?

As widely demonstrated in prior studies (*e.g.*, *Energy Environ. Sci.*, **2025**, 18, 1214-1231; *Nat. Catal.*, **2018**, 1, 508-515), IrO₂ exhibits superior electrochemical stability compared to IrO_x due to its more robust crystalline framework. IrO_x, with its defect-rich and loosely coordinated structure, is prone to hydration and Ir dissolution. The corresponding supplements has been incorporated into the Introduction section.

Our response to the reviewer's question (2): Is the support playing a role here other than easing the transformation? How does this agree with the claim of "... through strong metal-support interactions." on page 8 of the manuscript?

We acknowledge that the phrase "through strong metal-support interactions" on page 8 was imprecise and have removed it. While **Figure 2** shows an apparent reduction in dissolution for supported Ir/TiO_x@Ti compared to unsupported Ir NPs, the explanation for this phenomenon lies in **Figure 5**, where we clarify that the reduced dissolution is primarily attributed to the phase transformation. To substantiate this claim, we supplemented our analysis with electrochemical dissolution data for commercial crystalline rutile IrO₂, presented in Supplementary Figure 19. The results demonstrate that Ir/TiO_x@Ti exhibits dissolution comparable to commercial IrO₂, confirming their similar structural stability.

Our response to the reviewer's question (3): Also, the LOM/AEM crossover already occurs before reaching IrO. How is this reconciled with the higher stability for the completely transformed structure?

In the updated manuscript, we have corrected a previous statement regarding the catalytic mechanism. The reaction mechanism is not a "switch from LOM-dominated to AEM" but rather a "shift from a LOM-participated pathway to a complete AEM". To clarify, during the transformation of Ir to IrO₂ on TiO_x@Ti, LOM participation is progressively suppressed until it ceases entirely after structural stabilization. However, experimentally quantifying the precise

oxygen content and localization during this transformation remains challenging. To qualitatively describe this process, we constructed a simplified theoretical model where oxygen incorporation is limited to the subsurface layers of an Ir(110) slab model. Our focus is not on the absolute crossover values of LOM and AEM in **Figure 6e**, but rather on demonstrating that LOM becomes energetically unfavorable with increasing oxygen content. Therefore, this observation does not conflict with the fact that the fully transformed structure (with the highest oxygen content) exhibits enhanced stability.

3) Does the transformation to IrO₂ in any way depend on the size of the Ir particles? Can particles of any size be completely transformed?

Response 3: We appreciate the reviewer's insightful question. We acknowledge that Ir particle size might influence the transformation. However, in our synthesis system, Ir particle size could not be systematically controlled. In the Methods section, we describe the synthesis conditions as 180 °C for 3 hours. Despite varying reaction temperatures (up to 200 °C, exceeding ethylene glycol's boiling point of 197 °C, as showed in Supplementary Figure R1a) or extending reaction times (up to 5 h, as showed in Supplementary Figure R1b), Ir particle size nearly remained unchanged. This makes it challenging to systematically evaluate the impact of particle size on Ir transformation, which requires improved synthetic design in future work. Therefore, subsequent research will investigate the effects of different particle sizes by refining synthesis methods. Nevertheless, we emphasize that particle size is not the determining factor for the distinct transformation pathways between Ir/TiO_x@Ti and unsupported Ir NPs. We have included the size distribution of unsupported Ir NPs (Supplementary Figure 9), showing a mean diameter of 2.1 nm, nearly identical to the 2.0 nm of Ir in Ir/TiO_x@Ti. This confirms that the support plays the dominant role in driving the transformation.

Supplementary Figure R1. TEM images of Ir/TiO_x@Ti under different synthesis conditions: (a) 200 °C, 3 h; (b) 180 °C, 5 h.

4) Could the authors detail how the IrO_{0.25}, IrO_{0.5} and IrO models used for DFT analysis were constructed? How can the structural relevance of the contained active sites be ascertained?

Response 4: The model diagrams have been added to Supplementary Figure 26 in the revised manuscript, with detailed construction procedures incorporated into the Methods section. These models were established to elucidate the experimentally observed transition from LOM-participated to fully AEM-dominated OER mechanisms in our DEMS studies. This experimental phenomenon indicates that varying oxygen content within the structure indeed drives the mechanistic evolution. Through our progressively oxygen-enriched subsurface models, this observation finds rational thermodynamic interpretation: the initial LOM participation implies co-activation of limited lattice oxygen and Ir sites, whereas during the structural transformation, oxygen saturation renders lattice oxygen inert while exclusively activating Ir sites, culminating in complete AEM. This originates from the strengthened Ir-O structural framework induced by increased oxygen content, which reduces lattice oxygen accessibility for reaction participation and consequently suppresses LOM.

5) It is not entirely clear from the data how to authors arrive at the (211) surface orientation from the Fourier transforms.

Response 5: The identification of the (211) exposed facet was achieved through zone axis indexing methodology, which integrates high-resolution lattice imaging, FFT analysis, and crystallographic zone law. The procedural workflow comprises three key phases: (1) Cross-lattice fringe regions were identified in HRTEM images by locating intersections of three distinct lattice fringes, with their spacings and interplanar angles measured. (2) FFT analysis was applied to these regions to precisely map diffraction spots, which were then compared to theoretical lattice projections. (3) The zone axis (exposed facet) was determined using the zone law, calculated by cross-multiplying the Miller indices of adjacent lattice planes. The corresponding description has been added to Supplementary Figure 17.

6) For the DFT calculations, what semi-core states were included for the metal atoms? Is the k -sampling sufficient for metals? From where are the ZPE and entropy corrections taken?

Response 6: Our response to the reviewer's question (1): For the DFT calculations, what semi-core states were included for the metal atoms?

For the DFT calculations, the semi-core $3p$ states of Ti were explicitly included using the Ti_sv pseudopotential, while no semi-core states were included for Ir, with only valence electrons treated explicitly *via* the standard Ir pseudopotential.

Our response to the reviewer's question (2): Is the k -sampling sufficient for metals?

We apologize for the earlier error in reporting the k -sampling interval as $0.04 \times 2\pi \text{ \AA}^{-1}$. The corrected value used in our calculations is $0.03 \times 2\pi \text{ \AA}^{-1}$, corresponding to a denser k -point grid. This ensures sufficient accuracy for structural optimization in Ir systems (*J. Am. Chem. Soc.*, **2019**, *141*, 5409-5414).

Our response to the reviewer's question (3): From where are the ZPE and entropy corrections taken?

ZPE corrections for slab models were derived from vibrational frequency calculations, with entropy corrections for solids neglected. For gas-phase molecules (*e.g.*, H₂, H₂O), ZPE and entropy corrections were adopted from the widely recognized methodology developed by Nørskov and colleagues (*J. Phys. Chem. B*, **2004**,*108*, 17886-17892).

7) In figure S4, how is the energy difference defined? Does smaller imply an easier migration into the lattice?

Response 7: The energy difference is defined as the total energy difference between the optimized configurations before and after oxygen migration, which can be termed the oxygen adsorption energy difference or oxygen binding energy difference (*Phys. Rev. B: Condens. Matter*, **2008**, *78*, 045436). A smaller energy difference indicates that oxygen migration into the subsurface is thermodynamically more favorable, reflecting easier incorporation into the lattice. The corresponding descriptions have been incorporated into Supplementary Figure 4.

吉林大学 化学学院

Department of Chemistry

Jilin University

Changchun 130012, China

Dr. Prof. Xiaoxin Zou

State Key Lab. Inorg. Synth. & Prep. Chem.

Tel: +86-431-85168221

xxzou@jlu.edu.cn

<http://zouxxgroup.com/>

Reviewer's Comments and Our Responses:

Reviewer #5:

Electrochemical water splitting has become an attractive alternative clean and efficient hydrogen production method. To this end, the fabrication of low-cost, stable and high-efficiency electrocatalysts for water electrolysis is crucial. In this paper, the authors showed the fabrication of Ir/TiO_x@Ti catalyst for OER with high activity and high durability in acidic media. However, the novel opinions are difficult to be found. The following questions should be addressed before publication in other journal.

Response: We acknowledge your focus on novelty. We emphasize that the core innovation of our work lies in overturning the traditional understanding of reconstruction chemistry in Ir-catalyzed OER processes.

For nearly half a century, it is understood that electrochemical oxidation of Ir nanoparticles during OER is confined to their surfaces, forming a thin amorphous IrO_x shell. Our research fundamentally challenges this view. We demonstrate that supported Ir nanoparticles (dispersed on shell@core TiO_x@Ti particles) undergo a bulk phase transition from metallic Ir to crystalline rutile IrO₂, contrasting with the typical surface-limited amorphous IrO_x formation.

The key highlights and contributions are:

Extraordinary Phase Transition and Catalytic Mechanism Evolution. We report a unique case where supported Ir nanoparticles transform entirely into crystalline rutile IrO₂, bypassing the conventional thin amorphous IrO_x surface layer. This bulk phase transition drives a shift in the OER mechanism from the lattice-oxygen-participated mechanism (LOM) to the complete adsorbate evolution mechanism (AEM). This was elucidated through an integrated operando study, combining spectroscopic, electrochemical, and computational approaches.

Simultaneously Enhanced Activity and Stability. The reconstructed Ir nanoparticles form stable, (211)-oriented rutile IrO₂ nanocrystallites. These structures feature highly active Ir sites for OER while exhibiting remarkably low iridium dissolution and exceptional long-term stability (>1700 h). Crucially, a PEMWE employing the Ir/TiO_x@Ti anode achieves a high current density of 4.0 A cm⁻² at 2.0 V with a very low Ir loading of 0.3 mg cm⁻², representing a significant advance towards practical supported Ir catalysts.

In conclusion, this work fundamentally revises the established understanding of reconstruction chemistry in Ir-catalyzed OER. It provides critical new insights for designing advanced catalysts through directional control over catalyst reconstruction.

1. For TiO_x@Ti, what is the value of x? What impact does x value on the catalytic performance of Ir?

Response 1: The notation TiO_x is used to denote the amorphous titanium oxide phase, distinguishing it from crystalline TiO_2 . Precisely determining the stoichiometric parameter x within the amorphous TiO_x layer remains challenging. Nevertheless, surface-sensitive XPS analysis of the Ti 2p region (**Supplementary Figure R1**) offers valuable insight. Compared to commercial TiO_2 , the Ti 2p peaks for $\text{TiO}_x@\text{Ti}$ exhibit a distinct 0.2 eV shift towards lower binding energies. This shift clearly indicates that x is slightly less than 2. Furthermore, Raman spectroscopy suggests the amorphous TiO_x shell consists of $\text{Ti}_3\text{O}_5/\text{Ti}_4\text{O}_7$ -like clusters, implying a potential x value ranging from 1.65 to 1.8. The corresponding discussion has been added to Supplementary Figure 5.

Due to the lack of reliable quantitative techniques for determining x within the $\text{Ir}/\text{TiO}_x@\text{Ti}$ composite structure, establishing a definitive relationship between the x value and the catalytic performance of Ir faces significant obstacles. To address this, ongoing work is focused on synthesizing various crystalline titanium oxides (including Ti_3O_5 , Ti_4O_7 and TiO_2) to systematically investigate the correlation between composition x and activity.

Supplementary Figure R1. Ti 2p XPS spectrum of $\text{TiO}_x@\text{Ti}$ and TiO_2 .

2. In $\text{TiO}_x@\text{Ti}$, is the thickness of amorphous TiO_x shell controllable? What are the effects of different TiO_x shell thicknesses on the catalytic performance of Ir? What is the optimal thickness of TiO_x shell?

Response 2: Our experimental findings demonstrate that the thickness of the amorphous TiO_x shell is not controllable under the synthesis conditions explored. The synthesis of the $\text{Ir}/\text{TiO}_x@\text{Ti}$ catalyst requires specific conditions—reacting the Ir precursor with the Ti substrate at 180–200 °C for 3–5 hours to ensure uniform Ir loading and formation of the active structure. Crucially, these necessary reaction conditions inherently produce the passivated TiO_x layer at its limiting thickness (~5 nm, as confirmed in Fig. 2d). This is evidenced by comparative experiments where pure nano-Ti powder was reacted in ethylene glycol alone for 2 hours and 5 hours (**Supplementary Figure R2**). Despite the differing reaction times, the resulting TiO_x shell thickness consistently reached ~5 nm. This consistency indicates that the oxidation process reaches a thermodynamic limit,

forming a passivating TiO_x layer that prevents further bulk oxidation of the underlying Ti metal. The corresponding discussion has been added to Supplementary Figure 6.

Therefore, because the TiO_x shell thickness is intrinsically constrained to this limiting value (~ 5 nm) by the passivation mechanism under the required synthesis conditions, it is not possible to experimentally vary the thickness. Consequently, evaluating the effects of different shell thicknesses on catalytic performance or determining an optimal thickness is not feasible.

Supplementary Figure R2. TEM images of nano-Ti powder after reaction in ethylene glycol at 180°C in air for (a) 2 hours and (b) 5 hours.

3. The authors think the amorphous TiO_x shell consists of $\text{Ti}_3\text{O}_5/\text{Ti}_4\text{O}_7$ -like clusters. Is the molar ratio of $\text{Ti}_3\text{O}_5/\text{Ti}_4\text{O}_7$ in the amorphous TiO_x shell controllable? What are the effects of different molar ratios of $\text{Ti}_3\text{O}_5/\text{Ti}_4\text{O}_7$ on the catalytic performance of Ir?

Response 3: Our characterization, particularly Raman spectroscopy (Supplementary Figure 5), reveals features characteristic of $\text{Ti}_3\text{O}_5/\text{Ti}_4\text{O}_7$ -like clusters within the amorphous TiO_x shell. However, the local atomic arrangements of Ti_3O_5 and Ti_4O_7 are highly similar, leading to nearly identical characteristic Raman peaks for these phases. This spectral overlap makes it experimentally impossible to distinguish between Ti_3O_5 and Ti_4O_7 clusters, or to quantify their relative proportions (molar ratio) within the amorphous shell. Consequently, precise control over the $\text{Ti}_3\text{O}_5/\text{Ti}_4\text{O}_7$ molar ratio in the amorphous TiO_x shell is not feasible. Given this fundamental limitation in characterization and control, investigating the specific effects of varying this molar ratio on the catalytic performance of Ir is not possible. To address this, ongoing work is focused on synthesizing crystalline Ti_3O_5 and Ti_4O_7 to investigate the correlation between structure and activity.

4. In Figure 3a and 3b, what is the percentage of Ir content? Is it mass percentage content or molar percentage content? Please indicate clearly.

Response 4: In Figures 3a and 3b, the “percentage of Ir content” refers to the mass percentage content. To avoid any misunderstanding, we have added clarification to the caption of Figure 3a.

5. For Ir/TiO_x@Ti catalysts, is Ir completely converted to crystalline IrO₂ during the catalytic process? Why haven't you discussed achieving the optimal ratio of IrO₂/Ir?

Response 5: Our analysis confirms that Ir is fully converted to crystalline IrO₂ after 10 hours of catalytic operation. This complete oxidation is facilitated by the promoting effect of the TiO_x@Ti support, as discussed in relation to Fig. 4 and Fig. 5d. Critically, the EXAFS *R*-space data for Ir/TiO_x@Ti after 10 hours of catalysis shows the complete absence of metallic Ir–Ir bonds (**Supplementary Figure R3** and **Supplementary Table R1**). This provides direct structural evidence for the full conversion of Ir to the oxide state (IrO₂). The corresponding discussion has been added to Supplementary Figure 19. Consequently, IrO₂, rather than any IrO₂/Ir, serves as the actual steady-state catalytically active phase, contributing to the high activity and stability. Therefore, discussing an optimal IrO₂/Ir ratio is not relevant.

Supplementary Figure R3. Ir L₃-edge EXAFS spectra in *R* space of Ir/TiO_x@Ti after 10 hours of catalysis.

Supplementary Table R1. EXAFS fitting parameters at the Ir L₃-edge for Ir/TiO_x@Ti after 10 hours of catalysis.

Sample	Shell	CN ^a	R (Å) ^b	σ ² (Å ² ·10 ⁻³) ^c	Δ E ₀ (eV) ^d	R factor
Ir/TiO _x @Ti-OER	Ir–O	6.13±0.38	2.12±0.05	0.012±0.008	8.27±1.38	0.0038
	Ir–O–Ir	1.12±1.11	3.07±0.10	0.025±0.004		

Note: ^aCN, coordination number; ^b*R*, the distance between absorber and backscatter atoms; ^cσ², Debye-Waller factor to account for both thermal and structural disorders; ^dΔ*E*₀, inner potential correction; R factor indicates the goodness of the fit. *S*₀² is fixed to 0.815, according to the experimental EXAFS fit of Ir foil by fixing *CN* as the known crystallographic value. Fitting range:

$k(\text{\AA}^{-1})$: 2.0–12.0; $R(\text{\AA})$: 1.0–3.5; fitting space: R space. A reasonable range of EXAFS fitting parameters: $0.800 < S_0^2 < 1.000$; $CN > 0$; $\sigma^2 > 0 \text{\AA}^2$; $|\Delta E_0| < 10 \text{ eV}$; R factor < 0.02 .